# Prerequisite for COVID-19 Prediction: A Review on Factors Affecting the Infection Rate

**DOI:** 10.3390/ijerph192012997

**Published:** 2022-10-11

**Authors:** Shirley Gee Hoon Tang, Muhamad Haziq Hasnul Hadi, Siti Rosilah Arsad, Pin Jern Ker, Santhi Ramanathan, Nayli Aliah Mohd Afandi, Madihah Mohd Afzal, Mei Wyin Yaw, Prajindra Sankar Krishnan, Chai Phing Chen, Sieh Kiong Tiong

**Affiliations:** 1Center for Toxicology and Health Risk Studies (CORE), Faculty of Health Sciences, Universiti Kebangsaan Malaysia, Jalan Raja Muda Abdul Aziz, Kuala Lumpur 50300, Malaysia; 2Institute of Sustainable Energy, Department of Electrical & Electronics, Universiti Tenaga Nasional, Kajang 43000, Malaysia; 3Faculty of Business, Multimedia University, Jalan Ayer Keroh Lama, Malacca 75450, Malaysia

**Keywords:** COVID-19, face mask, infection rate, meteorological factor, physical distance, socioeconomic factor, vaccination

## Abstract

Since the year 2020, coronavirus disease 2019 (COVID-19) has emerged as the dominant topic of discussion in the public and research domains. Intensive research has been carried out on several aspects of COVID-19, including vaccines, its transmission mechanism, detection of COVID-19 infection, and its infection rate and factors. The awareness of the public related to the COVID-19 infection factors enables the public to adhere to the standard operating procedures, while a full elucidation on the correlation of different factors to the infection rate facilitates effective measures to minimize the risk of COVID-19 infection by policy makers and enforcers. Hence, this paper aims to provide a comprehensive and analytical review of different factors affecting the COVID-19 infection rate. Furthermore, this review analyses factors which directly and indirectly affect the COVID-19 infection risk, such as physical distance, ventilation, face masks, meteorological factor, socioeconomic factor, vaccination, host factor, SARS-CoV-2 variants, and the availability of COVID-19 testing. Critical analysis was performed for the different factors by providing quantitative and qualitative studies. Lastly, the challenges of correlating each infection risk factor to the predicted risk of COVID-19 infection are discussed, and recommendations for further research works and interventions are outlined.

## 1. Introduction

According to the Centers for Disease Control and Prevention (CDC), infection by severe acute respiratory syndrome coronavirus 2 (SARS-CoV-2) was the leading cause of death in the United States of America for the first time in 2020 with a total of 350,831 deaths in the same year [1]. The coronavirus was declared a global pandemic in less than two months after the World Health Organization (WHO) declared the virus outbreak as a Public Health Emergency of International Concern (PHEIC) in January 2020. As of 27 July 2022, the WHO has recorded that SARS-CoV-2 has infected over 568 million people, with cumulative deaths of over 6 million worldwide [2]. The cumulative number of people infected and the cumulative deaths by the virus are still increasing daily, even with high vaccine rollout and strict safety measures. SARS-CoV-2, also known as coronavirus disease 2019 (COVID-19), as officially named by the WHO, was first discovered in Wuhan, China [3]. The virus is claimed to be originated from bats through animal-to-human transmission, as previous studies have shown that some bat SARS-CoVs can potentially infect humans [4,5]. Bats are known coronavirus carriers, and researchers discovered that the genome of SARS-CoV-2 is similar to that of RATG13, a coronavirus first found in a horseshoe bat (*Rhinolophus affinis*) in the southern Chinese province of Yunnan in 2013. However, the genome of RATG13 is only 96% identical to that of SARS-CoV-2, implying that a closer relative of the virus—the one that infects humans—remains unknown [6]. Despite the plausibility of the bat-origin scenario, the search for a potential intermediate host is still ongoing, as SARS-CoV-2 samples were not obtained in nature after extensive and broad animal investigation [7,8,9].

In addition, infection of the virus for human-to-human transmission is mainly from respiratory droplets and direct contact [10]. The most common symptoms for symptomatic infected persons are fever, dry cough, and fatigue; upper respiratory tract symptoms can include pharyngalgia, headaches, and myalgia [11]. Although the symptoms seem to be like a normal fever and flu, complications such as acute respiratory distress syndrome (ARDS), respiratory failure, liver injury, acute myocardial injury, acute kidney injury, septic shock, and even multiple organ failure can occur for patients with other health complications including those with obesity, diabetes, and hypertension [12]. However, a person that is infected but is asymptomatic may cause a transmission source that could not be identified straightaway. For example, SARS-CoV-2/COVID-19 appears to be more infectious than SARS-CoV, whose transmission is mainly restricted to symptomatic patients, which may be due to the unidentified number of the asymptomatic persons. Over 600 million confirmed cases and over 6.4 million deaths have been reported worldwide as of 4 September 2022 [13]. According to current data, 80% of COVID-19 patients either have mild-to-moderate acute COVID-19 symptoms or are asymptomatic, 15% of people are expected to develop more severe diseases requiring hospitalizations, and 5% are critical infections requiring ventilation [14]. The WHO estimated about 10–20% of COVID-19-infected individuals, equating to approximately millions of people world-wide, experience persistent or new symptoms for weeks, months, or even years, irrespective of the viral status; this is defined as “long COVID”, “Post COVID syndrome”, or “Long Haulers” [15,16]. Long COVID can affect the functions of various human systems including the respiratory, gastrointestinal, cardiovascular, musculoskeletal, integumentary, and neurological systems. Long COVID symptoms include dyspnoea, fatigue, sore throat, persistent cough, cognitive impairment, headaches, heart irregularities, sleep difficulties, skin rashes, muscle soreness, concentration issues, and post-traumatic stress disorder symptoms [17]. The long COVID symptoms have impaired the capacity of COVID-19 survivors to function, work, and live a decent life [18,19].

Lack of awareness can result in serious outcomes in relation to COVID-19 and its complications. It is predominant for the public to be well informed about the cause of COVID-19 infection mainly because they are the principal front liner in the community. Increasing the public knowledge and understanding of the nature of the virus and its symptoms would lead to a lower infection rate in the community. This includes preventive measures and guidelines on how to take care of oneself and the surrounding populace. It is also critical for policymakers to be aware of these factors that cause COVID-19 infection because they are the major stakeholders in the policy process of revision. Enforcing movement restrictions orders, public curfews, and face mask wearing are parts of the policy that have been imposed in many countries to curb the spreading of COVID-19. The health practitioner is another key group that should be well informed on the infection factors of COVID-19 as they are the first in the line to ensure that people infected by the virus recover from it. Therefore, it is crucial to increase awareness of the infection factors to control the spreading of the virus and enhance future research on identifying new methods to curb COVID-19.

Despite the importance of awareness and understanding of the infection factors, there is currently no comprehensive review of the infection factors that cover thorough factors for COVID-19 virus infection. However, there are a few reviews reported on the infection rate through different facemask material [20,21,22], social distancing [23,24,25], and efficacy of different types of vaccines [26]. Therefore, this review will highlight various infection factors for COVID-19 and its challenges and recommendations. A comprehensive discussion on the different infection factors has been presented here. Furthermore, the correlation between the infection factors on the infection rate, the issues, and challenges related to the factors, and the recommendations for future research and interventions to reduce the transmission and infection have been thoroughly discussed and presented. This is not only to create awareness of the importance of understanding how infection factors of COVID-19 relate to each other and do not solely depend on one factor that affects the infection rate, but also to contribute to future research ideas to control COVID-19 and other airborne diseases. 

## 2. Face Mask Design and Material

Face masks have been frequently utilized since the pandemic as one of the strategies to stop the virus from spreading through respiratory particles in the air. Until now, the COVID-19 pandemic was still controlled by using face masks throughout East Asia, as well as for other respiratory infections such as influenza [27]. However, COVID-19 is still transmissible by both asymptomatic and symptomatic infected persons. Despite wearing a face mask, the risk of infecting others or being infected still exists, as Wang et al. found that it is only 79% efficient at preventing transmission [28]. However, this study does not clearly state what type of face mask was being used in the investigation. Another study by Maged et al. utilized a Susceptible, Exposed, Infectious, Recovered (SEIR) model to evaluate the impact of facemask wearing on the community spread of COVID-19. Results found that using face masks could reduce the reproduction number by 49% [29]. The same model also shows that face mask wearing could reduce the estimated peak number of infectious cases by at least 45% in the US and several countries in Europe [29]. 

However, different face mask materials have different levels of protection as different particles can pass through the face mask. Face masks can be classified into three categories: surgical face masks, respirator face masks, and cloth face masks [28]. The details of different types of face masks are summarized in Table 1. The bacterial filtration efficiency (BFE%) is a measurement to measure the resistance of a material and component against aerosol from coughing, speech, and sneezing.

It was found that N95 face mask is more effective than surgical face mask in preventing air leakage during cough. However, neither would entirely prevent the side leakage [30,31]. A comprehensive investigation on different materials and designs of face masks that affect the airflow through the mask and airflow through the mask leaks has been conducted by Chiera et al. [20]. Eight different types of masks, consisting of community masks (CM) and surgical masks (SM) were used in this experiment. In addition, another two commercially available filtering face piece (FP) respirator models were also used in the study, as reference for minimal airflow leakage as they have the ideal face fit design. Table 2 shows the face mask specifications used in the study. The total mask outward efficiency (TFE) was used to indicate overall face mask performance, which combined leakage efficiency with the BFE. The BFE of the face respirators FP-1 and FP-2 was 90 and >99%, respectively.

The surgical masks met their requirements, with BFE values greater than 98%. The BFE range for the community masks was 90 to 96%. Interestingly, it was found that highly breathable community masks (CM-3 and CM-4) have higher TFE values than surgical masks analysed in this study. It is worth noting that their findings show that a highly breathable community mask (e.g., CM-3) outperforms a commercially available face piece respirator (e.g., FP-1) in terms of TFE at medium and high flow rates. The TFE of the FP-2 mask was never surpassed by the masks examined in this study, due to the optimal flow repartition of that mask model. Indeed, BFE performance has a negligible effect on TFE, resulting in minor changes to the ranking of the tested masks. The study also revealed that a significant portion of exhaled air was not filtered by the mask but leaked unfiltered through the facial seal. The proportion of airflow passing through the mask filter and that leaking through the face seal was strictly related to the mask’s design, which included several factors including mask size, nose piece type, lace tension, filtering area size, filtering material composition and folding, as well as the filter’s breathability and the mask’s fit on the user’s face. Results demonstrated the importance of filter breathability in ensuring the face mask’s high TFE. Chiera et al. [31] suggest selecting highly breathable filtering materials for producing and using face masks to minimize air leakage and maximize user comfort. Therefore, it can be concluded that a high value of TFE can contribute in reducing transmission and infections, although none of the face masks provide 100% filtration efficiency. In addition to following the instructions for use to wear the face mask appropriately, it is recommended to adjust the nose piece shape and lace tension to improve face fit and establish a balance between user comfort and mask filtration efficiency. During the COVID-19 pandemic, an effective mask that is worn correctly is of utmost importance for preventing disease transmission. Other measures such as physical distancing, practicing good hygiene, and disinfecting, must also be followed in combination to reduce the transmission of the virus.

Furthermore, the WHO stated that wearing a face mask from a breathable material properly will not cause health problems [32]. During the COVID-19 pandemic, many countries have imposed the rule of wearing face mask to curb the spread of COVID-19. However, there is a group of people who are facing difficulties when wearing a face mask and require exemptions from wearing face masks. The group consists of children younger than 2 years old [33], people with disabilities such as hearing [34], breathing [35], and people with disabilities as defined by the American with Disabilities Act (ADA) [36]. Children younger than 2 years old are not recommended to wear any type of mask as they are still having very small airways that can make them struggle to breathe. In addition, they need assistance when removing the mask, and without it, risk of suffocation is very high [35]. One difficulty faced by person with hearing disabilities is that the loop of the face mask may pose challenges for people who wears hearing aids as it can cause the device to fall off [34]. People with disabilities are already at high risk of serious illness due to COVID-19, and they have experienced overwhelming stigma and isolation during the pandemic. Exemption on the face mask mandatory order has been raised by the group and their caregiver. There are several practical considerations provided by CDC for mask exemption and the exemption must come from the authorized physician [37]. Table 3 shows the subset of persons with disabilities that are either exempted or may be exempted under the CDC’s requirement to wear face mask.

Due to the exemptions of face masks for these groups can lead to spike in COVID-19 infections. Furthermore, a study predicted that this exemption may lead to significant risk of COVID-19 for many people, especially mentally ill patients [38]. This prediction was proven by Wang et al. [39] where people who have been diagnosed with mental illness and who are exempted from wearing face masks, are much more likely to contract COVID-19 than those without mental illness who wear face masks. This group of people also shows worse outcomes when infected, as shown by higher hospitalization rates and death.

## 3. Meteorological Factors

Meteorological factors may play an important role in the spread of a virus [40,41]. Before the emergence of COVID-19, the impact of meteorological conditions on the development of other infectious diseases was also examined [42,43]. Climate factors such as temperature, humidity, wind speed, dew/frost point, precipitation, and surface pressure can influence droplet stability and consequently COVID-19 transmission [44,45,46]. It has been documented that temperature can affect the life cycle and replication of viruses, while humidity and wind speed can impact the transmission of airborne viruses such as coronaviruses (CoVs) [43,47,48]. Generally, CoVs are enveloped positive-sense RNA viruses of both medical and veterinary importance [49]. The envelope structures of SARS-CoV-2 are susceptible to physical and chemical conditions and can be deactivated or damaged by extreme pH, heat or ultraviolet light (UV) [50]. The outermost structural protein of SARS-CoV-2 is characterized by “Club-like spike protein” which exhibited active and inactive states at different temperatures [51]. Thus, regions with lower temperatures are more susceptible to infection than regions with higher temperatures [52]. The number of COVID-19 cases toward the Earth’s poles as latitude increased [50]. Consequently, COVID-19 peaks occur in the winter, manifesting as local epidemics lasting a few weeks or months [53,54,55]. 

Several studies reported that the infectivity of COVID-19 is affected by temperature. A machine learning case study in Spain has found an inverse correlation between temperature and the daily number of infections. They found a decrease of approximately 200 daily infected people per 1 °C increase in average temperature [56]. Prata et al. explored the relationship between annual average temperatures (16.8 °C to 27.4 °C) and confirmed COVID-19 cases in (sub)tropical cities of Brazil [57]. The findings of the study revealed that temperatures had a significant negative linear relationship with the number of confirmed cases. They demonstrated that each 1 °C rise was associated with a −4.8951% (t = −2.29, *p* = 0.0226) decrease in the number of daily cumulative confirmed cases of COVID-19. Similarly, Wang et al. studied the effect of temperature on the number of confirmed cases in 429 cities in the world from 20 January 20 to 4 February 2020 [58]. The average, minimum, and maximum temperatures were 8.72 °C, 6.70 °C, and 12.42 °C, respectively. In the case of lower temperatures, every 1 °C rise in average, minimum, and maximum temperatures increased the total number of cases by 0.83, 0.82, and 0.83, respectively. They also estimated that each 1 °C increase in the minimum temperature decreases the cumulative number of cases by 0.86. These results were supported by a laboratory study that reported increasing temperature and relative humidity (RH) accelerates the inactivation of SARS-CoV-2 on surfaces. When the temperature was raised from 24 to 35 °C, the virus decayed quicker and had a shorter half-life at 20% RH (t_1/2_ = 7.33 ± 1.33), 40% RH (t_1/2_ = 7.52 ± 1.22), and 60% RH (t_1/2_ = 2.26 ± 1.42). Depending on the RH, the viral half-life ranged from 6.3 to 18.6 h at room temperature (24 °C) but decreased to 1.0 to 8.8 h when the temperature was elevated to 35 °C [59]. A further study performed by Biryukov et al. indicated that at a temperature of 54.5 °C, the virus half-life ranged from 10.8 ± 3.0 min and the infectivity decreased by about 90% at 35.4 ± 9.0 min, indicating that in an environment that can reach such high temperatures, such as the interior cabins of parked vehicles, the infectivity decreased in a significant way [60]. However, a time series analysis performed by He et al. to investigate the impacts of temperature and RH on daily COVID-19 confirmed positive cases in different Asian cities reported that daily new confirmed cases of COVID-19 were more correlated with average temperature than RH. In Beijing (*r* = −0.565, *p* < 0.001), Shanghai (*r* = −0.47, *p* < 0.001), and Guangzhou (r = −0.53, *p* < 0.001), daily new confirmed cases were negatively correlated with average temperature [61]. In Japan, however, there was a positive association (*r* = 0.416, *p* < 0.001). Generalized additive modelling (GAM) analysis revealed that the number of daily new confirmed cases positively correlated with both average temperature and RH in the majority of cities (Shanghai, Guangzhou, Hong Kong, Seoul, Tokyo, and Kuala Lumpur). 

Another recent study conducted by Liu et al. found that diurnal temperature (DTR), ambient temperature (AH), and absolute humidity (AH) are significant attributes with negative correlations to confirmed case counts [62]. Their study revealed that each 1 °C increase in DTR and AT was associated with the decline of daily confirmed case counts, and the corresponding pooled relative risks (RRs) were 0.90 (95% CI: 0.86, 0.95) and 0.80 (95% CI: 0.75, 0.85), respectively. Their analysis also showed that every 1 g/m^3^ increase in AH was significantly associated with reduced confirmed case counts in lag 07 and lag 014, and the pooled RR was 0.72 (95% CI: 0.59, 0.89) and 0.33 (95% CI: 0.21, 0.54), respectively. One possible reason is that low AH increases coronavirus stability and promotes transmission, as influenza did [63]. Hence, they concluded that the COVID-19 infectivity is favoured by weather with low temperature, mild diurnal temperature range, and low humidity. The low temperature such as in winter weakened the humans’ innate immunity. The low temperature decreased blood supply and, consequently, reduced immune cells’ supply to the nasal mucosa [63]. Furthermore, low humidity diminishes the ability of airway cilia cells to remove virus particles, secrete mucus, and repair airway cells, so exposing the host to the virus [63,64]. Cells release signal proteins during viral infections to warn neighbouring cells of the threat of virus invasion. However, this innate immune defence system is impaired in low-humidity environments [64,65]. Thus, the human body is more susceptible to viral infection in low temperature and low humidity environments [66]. 

A study was conducted in Jakarta, Indonesia to investigate the relationship between temperature, humidity, wind speed, sunshine hours, and rainfall on COVID-19 cases. The findings of the study demonstrated that low wind speed, temperature, and sunshine hours were significantly correlated with higher COVID-19 cases (*r* = −0.314; *p* < 0.05, *r* = −0.447; *p* < 0.01, *r* = −0.362; *p* < 0.05, respectively) [67]. This study also concluded that low wind speed contributes to an increase in COVID-19 incidence. This finding is consistent with those reported by Bashir et al., who demonstrated that wind speed is an environmental factor contributing to rising COVID-19 cases in New York, NY, USA [68]. An ecological study also showed that days with temperatures ranging from 16 to 28 °C and slow average wind speeds of <8.85 km per hour (KPH) showed higher COVID-19 cases (aIRR = 1.45, 95% CI = [1.28–1.64], *p* <0.001) than days with average wind speeds of 8.85 KPH [69]. Coccia revealed that low wind speed combined with high levels of air pollution may promote a longer persistence of viral particles in polluted urban air, thus favouring an indirect mode of transmission of SARS-CoV-2 in addition to the direct transmission through human-to-human transmission dynamics [70]. Another study indicated that the COVID-19 infection rate was inversely correlated (*p* < 0.05) to UV radiation and temperature, indicating a possible role of sunlight in decreasing SARS-CoV-2 infectivity [71]. According to Whittemore, direct skin exposure to sunlight promotes vitamin D production, a vital component that regulates the immune system [72]. Vitamin D can lower the risk of respiratory tract infections such as COVID-19 through a multitude of cellular interactions that involve the maintenance of endothelium integrity, a reduction in the production of inflammatory cytokines and elevation of angiotensin-converting enzyme 2 (ACE2) [73]. 

However, a recent study indicated that digital elevation model (DEM) (*β* = 0.361, *p* = 0.012), land surface temperature (LST) (*β* = 0.236, *p* = 0.040), rainfall (*β* = 0.439, *p* = 0.017) and windspeed (*β* = 0.212, *p* = 0.038) were positively correlated with the COVID-19 infection rates, while wind pressure had a negative correlation (*β* = −0.260, *p* = 0.015). These results mean that a 1 m increase in DEM, 1 °C increase in LST, 1 mm increase in rainfall, and 1 mph increase in wind speed significantly increased the COVID-19 infection rate by 0.36, 0.24, 0.44, and 0.21, respectively, while 1 ms^−1^ increase in wind pressure led to a decrease of 0.26 in COVID-19 infection rates. All these five meteorological attributes were highly significant with an *R*^2^ value of 0.73 which inferred that the 73% variations in COVID-19 infection rate can be explained by these parameters [74]. Gupta et al. found that latitude (*β* = 5.689; *p* < 0.001), longitude (*β* = 2.589, *p* < 0.001), and humidity (*β* = 14.723, *p* < 0.001) were positively correlated with COVID-19 infection rates; however, hours of sunlight (*β* = −17.803, *p* = 0.090), temperature (*β* = −8.938, *p* = 0.011), and windspeed (*β* = −7.492, *p* = 0.033) showed a negative correlation with COVID-19 infection rates [75]. These results implied that a unit increment in latitude, longitude, and humidity increases the COVID-19 infection rates by 5.689, 2.589, and 14.723, respectively. The increase of a single unit in hours of sunlight, temperature, and windspeed decreases the COVID-19 infection rates by 17.803, 8.938, and 7.492, respectively. Another ecological study was conducted by Sarmadi et al. on 406 authorities of the UK and revealed that a significant (*p* < 0.01) negative correlation between average high temperature (*r* = −0.33), average mean temperature (*r* = −0.22), average dew point (*r* = −0.29) and COVID-19 incidence rates [76]. However, a significant (*p* < 0.01) positive correlation was found between average wind speed (*r* = 0.192) and COVID-19 incidence rates in the study. Hence, they concluded that long-term high temperature and dew point are the most significant in decreasing the COVID-19 incidence rates, but wind speed increases the rates. Different studies indicated contradictory impacts of the meteorological variables on the infectivity of COVID-19. It is undeniable that some of these findings were influenced by the countries chosen for the study, the methodology of analysis used, and other confounding factors that affect the phenomenon that may not have been eliminated in some of these studies.

## 4. Distance

### 4.1. Physical Distance

Keeping a physical distance, commonly known as ‘social distancing’, is an essential step in preventing the spread of COVID-19, where physical interaction is reduced between people [77]. The World Health Organization (WHO) has recommended at least 1 m of physical distance to avoid COVID-19 infection [78]. In contrast, the Centers for Disease Control and Prevention (CDC) has mentioned that a physical distance of at least 6 feet (about 1.5 to 2 m) must be kept from others [79], even for healthy people, to reduce and slow the transmission of COVID-19. Social distancing also means avoiding gathering spaces such as schools, supermarkets, and public transportation. Consequently, if a person has been exposed to the coronavirus and becomes ill, they need to be isolated and quarantined to prevent them from spreading it to others.

The rapid spreading of the COVID-19 virus worldwide has increased the related research to understand the new coronavirus, SARS-CoV-2 and find the best ways to treat it. Preventive measures are among the studies conducted to investigate the early protection against COVID-19 before the vaccines are found and created. One of the preventive measures is to study the effectiveness of physical distancing in reducing SARS-CoV-2 transmission. The “physical distancing rules” have been implemented in many countries and successfully decreased the incidence of COVID-19 infections. The news from the Reuters website has listed a few countries that have recommended physical distancing of 1 m, including China, Denmark, France, Hong Kong, and Singapore. Other countries such as Australia, Belgium, Greece, Germany, Italy, Spain, and Portugal advise people to keep 1.5 m apart [80]. 

Even though the incidence of COVID-19 infection is decreased by maintaining the recommended distance in all countries, this issue has become a debate among scientists as the research found that SARS-CoV-2 is an airborne transmission due to its persistence in aerosol droplets in a viable and infectious form [81]. Airborne transmission is defined by the WHO as the spread of droplet nuclei—the residue of dried respiratory aerosols (≤5 µm in diameter) that results from evaporation of droplets coughed or sneezed into the atmosphere or by aerosolization of infective material [82]. A study has shown that by considering the moist and warm environment, the droplets containing SARS-CoV-2 may evaporate for much longer and can travel 23–27 feet (about 7–8 m) [83]. Therefore, the recommended distance of about 1–2 m is probably ineffective against COVID-19 infections, especially if the places already have infected people. 

A recent study has demonstrated the effect of physical distancing against COVID-19 infections differs in each country based on many factors [84]. The graph in Figure 1 (based on the study in [84]) shows that the country with a small population has reported the highest incidence of COVID-19 infections. Lack of public communication is one of the factors that contribute to these results. Generally, during the COVID-19 incidence, the government will update the necessary information regarding COVID-19 incidence and recommends the physical distancing rule based on changes in the COVID-19 situation. The communication policy that most governments have adopted to provide information on COVID-19 to the public is through daily press briefings that can be watched on daily TV updates or streamed live online. A few countries also use infographic materials to communicate with the public. Consequently, some countries with a small population with low internet services or lack communication facilities receive critical information quite later than countries with a good internet connection. Therefore, a simple and easy-to-understand communication strategy is necessary if the countries continue to adjust physical distancing approaches in response to rapidly changing COVID-19 circumstances.

To date, the effectiveness of distancing measures was one of the most debated as they have different impacts on the number of cases. This issue has drawn the modelers’ team’s attention, leading to the development of many mathematical models [85,86]. The models are undeniably helpful in several aspects, such as guiding policymakers about the development of the epidemic and healthcare demand. One critical event in adaptive policy making linked to evidence produced in mathematical models is the infection control strategy based on assumptions of protective community or ‘herd’ immunity through combined social distancing measures and restrictions on population movement [87]. In the UK, the model developed by MRC Center for Global Infectious Disease Analysis at Imperial College London in collaboration with the World Health Organization (WHO) is often cited as the reason for the government to turn its policies to strict social distancing and lockdown [88]. The significant difference in the death numbers with and without strict measures were 20,000 and 500,000, respectively, have been misinterpreted by the media as a drastic change in the model assumptions and has raised questions about its accuracy [89]. Generally, the mathematical model is synthesized, processing the available data, and providing a structured framework to understand the epidemical and social mechanisms behind the outbreak. Even though mathematical modelling is a powerful tool for understanding the transmission of COVID-19 and exploring different scenarios, they cannot be considered accurate prediction tools since these models lack thorough formal data validation [90,91].

### 4.2. Mobility Reduction

The physical distancing rule is not only limited to maintaining the distance between each other at 1–2 m, but also includes reduced mobility. The workers are encouraged to work from home, and the children are also not allowed to attend school. At certain times, the malls’ or supermarkets’ operations must stop and close during the pandemic. Research has shown that reduced mobility of the community has decreased the COVID-19 cases all over the country that practice on physical distancing rule. The analysis of 134 countries that have practiced the physical distancing rule with reduced mobility shows a reduction of 65% of the new COVID-19 cases [92]. A study in [93] has investigated the rate of COVID-19 infection during the mobility reduction in a community based on the age difference. Based on that study, we summarized their findings in Figure 2, which shows that children below 5 years old are less infected by COVID-19. In contrast, working-age adults aged 25 to <60 years old have the highest cases. 

According to the results from Figure 2, it is deduced that the transmission of SARS-CoV-2 is higher in the workplace where some industries are still operating during the pandemic, such as the factories, healthcare, and retail sectors. Despite following the physical distancing rule at the workplace, the possibility of higher transmission between each adult is possible, perhaps due to the poor ventilation of the working places. Asymptomatic infections are also known as potential sources of transmission for COVID-19. Ma et al. reported the percentage of asymptomatic SARS-CoV-2 infections among populations tested for and confirmed COVID-19 in their meta-analysis study [94]. They found the pooled percentage of asymptomatic infections was 0.25% among the tested population and 40.50% among the confirmed population. The potential transmission risk is higher when a high percentage of asymptomatic infections. The two factors such as physical distancing and asymptomatic infections are the reasons that caused the COVID-19 outbreak. Densely populated places always become the main factors that led to ‘super-spreader’ events of COVID-19 infections where the physical distancing between individuals is not possible. Moreover, the asymptomatic infections were responsible for the late detection and delayed isolation of cases leading to widespread outbreaks [95]. Therefore, it is essential to perform contact tracing, testing and isolation of such super-spreaders from the other members of the community to stop the spread of the COVID-19 pandemic.

Highly populated places also include mass gathering events where the concentration of people at a specific location is higher. Mass gathering events generally include indoor and outdoor places and are considered super-spreading events because of the higher number of people attending, which estimated greater risk of infections [96]. Additionally, SARS-CoV-2 transmission is higher in households or places that hold a small gathering of people. Whaley et al. have studied the rate of COVID-19 infection using small gatherings, specifically the celebration of birthday parties [97]. The study found that the COVID-19 diagnosis rate is significantly greater in those households two weeks after the birthday celebration. Other commonly crowded places are the nursing homes that were more likely to experience larger and deadlier COVID-19 outbreaks. Brown et al. have found a total of 5218 residents (6.6%) developed COVID-19 infection, and 1452 (1.8%) died of COVID-19 infection in Canadian nursing homes [98]. The outbreak of COVID-19 also occurred on the cruise ship, where the individual isolation of all aboard was impossible. The first cruise ship affected by the COVID-19 outbreak was the Diamond Princess from Japan, where 619 cases had been confirmed (16.7% of the population on board), including 537 passengers and 82 crew members [94]. Hongoh et al. also analysed the criteria for evaluating transmission risk for indoor and outdoor places such as gyms, nightclubs, weddings, and others [99]. Therefore, it is suggested that policy interventions to limit SARS-CoV-2 transmission should be employed in other crowded places to stop the COVID-19 outbreak as much as possible. Moreover, theoretically, the transmission of SARS-CoV-2 droplets is probably easier within adults with little height difference compared with children or short people. Figure 3 shows the height difference has an impact on the transmission of the droplet. The height difference might affect the transmission rate within the same length of distance between two people. This is because, at a parallel height, the spreading of the virus becomes more direct compared to a different height. 

Previous studies have shown that SARS-CoV-2 also be transmitted indirectly through fomites or surfaces [100,101]. Numerous researchers have studied how long SARS-CoV-2 can survive on various porous and non-porous surfaces [102,103,104]. A study has reported the inability to detect the viable virus within minutes to hours on porous surfaces, such as clothes. Meanwhile, on non-porous surfaces such as plastic and glass; the viable virus can be detected for days to weeks. Apparently, faster inactivation of SARS-CoV-2 on porous compared with non-porous surfaces might be caused by the rapid droplet spreading due to the capillary action and fibres present on the porous surface [105]. The studies on surface survival indicate that a 99% reduction in infectious SARS-CoV-2 and other coronaviruses can be expected under typical in-door environmental conditions within 3 days (72 h) on common non-porous surfaces such as stainless steel, plastic, and glass [106,107]. However, experimental conditions on both porous and non-porous surfaces do not necessarily reflect real-world conditions due to the factors that can degrade the virus, such as ventilation and changing environmental conditions [108]. 

In this section, we have reviewed the effect of distance as a preventive measure against COVID-19 incidence. Although physical distancing effectively reduces SARS-CoV-2 transmission, the government needs to consider the rules as there are objections to physical distancing rule, especially on reducing mobility. In addition, a study has reported that reduce mobility or physical distancing might lead to depression and anxiety in some people [109]. This issue has, of course, had an impact on social stability where it increased the psychological effects on the population, including mental health problems and domestic violence. Therefore, all forms of psychological support should be implemented to help all the individuals while following the physical distancing rules to stop the outbreak of COVID-19. 

## 5. Ventilation

Proper ventilation ensuring enough outdoor air comes indoors is a critical approach to lowering pollutants or contaminants, including SARS-CoV-2. In general, increasing ventilation helps reduce the risk of airborne transmission of SARS-CoV-2 and decreases COVID-19 infections. A simple and economical way to provide ventilation for a room is taking advantage of atmospheric pressure differentials such as wind pressure moving air sideways. For example, open doors and windows as much as possible to allow more fresh air to move inside. Moreover, use fans to move virus particles in the air from inside the room to outside. Recently, with the increased COVID-19 incidence, it was advised to consider using a portable air purifier with high-efficiency particulate air (HEPA) cleaner to provide air filtration [110]. However, these devices are costly and are in limited supply for certain countries.

Usually, the ventilation rates in public buildings such as offices, schools, elevators, and public transport are significantly lower for various reasons, including companies attempting to limit airflow for energy and cost savings reasons. Recently, a study was conducted to investigate sufficient natural ventilation rate in the COVID-19 situation in a school classroom [111]. School classrooms usually have poor ventilation because it is often kept close to avoid outside noise, and their occupancy density is very high. Therefore, the school classroom is easily exposed to virus infection. The study investigated how the ventilation rate by opening the classroom windows affected the COVID-19 infection probability. A carbon dioxide (CO_2_) tracer was used to quantify the ventilation rate of the classroom. They summarized the findings based on the study in Figure 4 by taking the average of 1 h of the subject teaching in a classroom and considering all the students are wearing a mask. The ventilation rate varies by the percentage of windows opening. Increasing the percentage of windows opening will decrease the concentration of CO_2_ in a classroom, hence improving the ventilation.

Recently, the CDC updated that COVID-19 has shifted toward endemicity when the Omicron variant of SARS-CoV-2 led to an enormous surge in positive cases across the US [112]. It means that COVID-19 is still around but at a level that is not causing significant disruption in our daily lives. Consequently, COVID-19 infection can be at high, low, or manageable levels where we do not have to close down schools or industries. Therefore, most countries focus on preventing virus transmission rather than reporting the percent of test positivity and COVID-19 cases daily. However, since the operation of industries, schools, and other activities will normally be operating every day, it is essential to ensure the working place or any buildings is adequately ventilated to reduce the transmission of the virus, along with other prevention factors [113]. 

Since we live in the endemic of COVID-19 that is spreading through society at an extremely fast pace, many academics and industry professionals are raising the question of whether the current ventilation strategies are outdated and inadequate for such contagious diseases. It appears that the fresh air supply is not enough to improve the ventilation system of a building. A review in [114] has discussed the ventilation strategies to reduce the risk of COVID-19 transmission in high occupancy buildings considering airflow dynamics through occupied spaces. The classification of the ventilation strategies by airflow dynamics is shown in Figure 5. 

In general, the risk of transmission is higher in indoor spaces than in outdoor spaces, where it might be harder to keep people apart, and there is less ventilation. Increasing ventilation also will benefit indoor air quality by reducing exposure to products used for cleaning and disinfecting potentially contaminated surfaces. However, improving ventilation with all or mostly outside air may not always be possible or practical. In addition, some ventilation strategies might not be adopted as it is costly. In such cases, limiting the number of people present in the building or specific rooms might be the best alternative way to increase the ventilation rate per person. Other preventive measures also need to be practiced with a good ventilation system, such as keeping a distance and wearing a mask to decrease virus transmission rate.

## 6. Vaccines

Since the outbreak of COVID-19, vaccine development has been remarkable, with numerous vaccines distributed worldwide. It was found that the vaccines have high effectiveness in the real-world setting compared to clinical trials [115]. The protection against severe infection or death in the overall populace was more than 60% and primarily close to 100%. Furthermore, the majority of the studies showed that the vaccines provided protection against infection, signifying the potential for “herd immunity” through reduced transmission [115]. 

However, the efficacy of COVID-19 vaccine is studied by many researchers and is diverse in epidemic settings across the world [115]. Consequently, several potential sources of heterogeneity between the study make comparing different vaccine efficacy and/or effectiveness difficult [116,117]. This section will focus on vaccine equity, vaccine hesitancy and efficacy that could compromise the global COVID-19 response [115].

### 6.1. Vaccine Equity

According to the recent data by the WHO, as of June 2022, only 58 out of 194 WHO’s Member States had reached the 70% of the vaccinated population, and just 37% of healthcare workers had completed the course of primary vaccination in low-income countries [118]. Vaccine equity implies that, regardless of a country’s economic standing, vaccines should be distributed to all countries depending on need. However, challenging political, economic, social, diplomatic, and health-related issues influence how vaccinations are distributed. As of July 2022, the WHO reported that 3 in 4 people (72.45%) in high-income countries had been vaccinated with at least one dose, while only 1 in 5 people (20.66%) in low-income countries have been vaccinated with at least one dose [2,119]. Through vaccination, the complication and risk after being infected by the virus can be lowered. It is also found that the vaccine can provide a high degree of protection and thus reduce infections [115]. However, based on the data recorded by the WHO, the number of cumulative cases per number of populations for high-income countries is higher compared to low-income countries. Nonetheless, since the vaccination rate for low-income countries is still low, the number of cumulative deaths per the number of cumulative cases by COVID-19 for low-income countries is higher (2.3%) compared to the high-income countries (0.74%) [119]. This shows that even with high vaccination rate, people can still be infected, but with a low risk of further health complications and death.

### 6.2. Vaccine Hesitancy

Vaccine hesitancy and acceptance among general public and healthcare practitioners shows a significant role in successfully controlling the COVID-19 pandemic. The increasing of vaccine hesitancy groups has led the WHO to identify this as a major threat to global health. In addition, the general public has access to health information from various sources, including the internet, that could lead to misinformation [120]. Solís Arce et al. reported that vaccine acceptance in low- and middle-income countries is higher than in high-income countries [121]. It was also found that the highest vaccine with more than 90% acceptance rates among the general public were from Ecuador, Malaysia, Indonesia and China. On the other hand, the lowest vaccine acceptance rates with less than 60% among the general public were from Kuwait, Jordan, Italy, Russia, Poland, the United States, and France [122]. COVID-19 vaccine hesitancy was the highest among people aged from 18 to 24, non-Asian people, and low educational background (≤high school diploma) adults with COVID-19 infection history [123]. 

According to research by The Clayman Institute for Gender Research, women were significantly more likely to express a desire to delay or reject the COVID-19 vaccine than men [124]. These findings are also consistent with the existing literature on vaccine hesitancy, where only 45.2% of women accepted vaccination compared with 54.8% of men [125]. Pregnant and breastfeeding women are among the sub-groups attributed to women’s higher percentage of vaccine hesitancy [126]. Saitoh et al. reported that pregnant women were hesitant to receive the vaccine because of safety and efficacy concerns and were more likely to fear adverse effects [127]. Vaccine hesitancy also includes the group of people that recovered from COVID-19 as shown in a study by Gerussi et al. [128]. The study found that 59.2% of the recovered people refused to accept the vaccine and more than half of them were female participants. This could be explained by (a) their negative recent disease experience, which is still burdening them; (b) a lack of clear public information on the SARS-CoV-2 vaccine; and (c) the belief that they are immune, as evidenced by surveys on vaccinal propensity among people after H1N1 [129,130]. In fact, there are six main reasons why people are hesitant about getting the vaccine: (i) concerned about the efficacity of the vaccine, (ii) safety and/or side effects of the vaccine, (iii) does not think they need the vaccine, (iv) do not trust the vaccine since it was developed so quickly, (v) do not trust the pharmaceutical companies and the government, or (vi) does not think that the COVID-19 is a big threat [121,131]. Low vaccine acceptance may lead to low vaccination rate and often precedes an infectious disease outbreak and increasing the number of infections [132,133]. Some studies investigating the efficacy of the COVID-19 vaccine found a reduction in transmission in vaccinated individuals infected by COVID-19 [134]. It was also reported that there were reductions in viral load in the respiratory tract in vaccinated persons compared to unvaccinated persons [135].

### 6.3. Vaccine Efficacy

Among the protection measures to control or end the COVID-19 pandemic, an extensive vaccination program plays the most critical role in controlling the highly transmissible and pathogenic SARS-CoV-2 infection. The vaccines for SARS-CoV-2 that shows safety, effectiveness, and cost-efficiency is generally thought to be the most promising intervention to eventually end the COVID-19 pandemic by establishing herd community among populations. The vaccines for COVID-19 were first investigated during the second waves of COVID-19 and available during the third waves at the end of 2020 [85]. Many vaccine candidates have been clinically investigated; however, only a few were accepted for vaccination against COVID-19 and approved by certain countries, as shown in Figure 6. Many countries highly accepted and approved the type of vaccines manufactured by AstraZeneca/Oxford and Pfizer/BioNTech. Other vaccines that had reached the required safety and efficacy standards approved by many countries also include Janssen (Johnson and Johnson), Sinopharm, Moderna, and Sinovac.

According to C. Chambers in her articles published by Healthy Debate on 11 March 2021, the Pfizer and Moderna vaccines have the highest efficacy at around 95%. Meanwhile, lower efficacy was found for AstraZeneca and Janssen, in the 60–70% range [136]. The effectiveness of the vaccination was measured by how many vaccinated people contract COVID-19 compared with how many infections occur in the control group. Contrarily, it represents the proportion of COVID-19 infections that vaccination could prevent. 

The studies on the vaccination are also continuing to investigate whether the vaccine shots can stop people from getting infected and passing on the SARS-CoV-2 virus. Accordingly, vaccines that prevent transmission could help to bring the pandemic under control if they are given to enough people. Preliminary analyses suggest that at least some vaccines are likely to have a transmission-blocking effect; however, it is difficult to confirm that effect according to Marc Lipsitch, an infectious-disease epidemiologist at the Harvard T.H. Chan School of Public Health in Boston, Massachusetts [137]. Levine-Tiefenbrun and his co-workers observed a significant drop in viral load in a small number of people infected with SARS-CoV-2 in the two to four weeks after receiving their first dose of the Pfizer vaccine compared with those who caught the virus in the first two weeks after the injection [138]. Meanwhile, an investigation from Ioannou et al. showed comparable viral loads among vaccinated and non-vaccinated health care workers infected by SARS-CoV-2 (B.1.1.7/Alpha variant), suggesting suboptimal protection of Pfizer vaccines against the Alpha variant of SARS-CoV-2 virus [139]. Hsu et al. studied the transmission risk of close contact with the SARS-CoV-2 Delta variant, where they found a significant reduction in transmission of more than 50% in vaccinated compared with the non-vaccinated infected person [140]. 

Regardless the effectiveness of the vaccine in preventing COVID-19 infection, it was found that the immunity of the person will drop after a certain period of time. A study of the duration of protection of COVID-19 vaccine effectiveness indicates that vaccine effectiveness decreases more against infection and symptomatic disease than against severe disease in the 6 months after full vaccination [141]. This decreasing vaccine efficacy or effectiveness is probably caused by waning immunity. The concern of the duration of the protection and SARS-CoV-2 variants has encouraged additional vaccine doses known as booster vaccines, which could help reduce the risk of disease transmission. This is especially critical during the Omicron surge, as this variant appears to cause infection even in persons with vaccine-induced immunity [142]. Abu-Raddad et al. showed Pfizer/Moderna (mRNA) boosters were highly effective against symptomatic Delta infection but less effective against symptomatic Omicron infection. However, both variants strongly protected mRNA boosters against COVID-19–related hospitalization and death [143]. An investigation by Tenforde et al. showed the vaccinated persons who develop breakthrough infection are much less likely to be hospitalized or to die from COVID-19, reflecting multiple aspects of adaptive immunity induced by immunization [144]. In conclusion, booster vaccine doses reduce the risk of infection and prevent transmission of SARS-CoV-2.

## 7. Socioeconomic Factors

It is crucial to recognize the key socioeconomic factors contributing to COVID-19 infection risk and rate to expedite mitigating efforts and measures.

### 7.1. Population Density

The World Bank defines population density as midyear population divided by land area in square kilometres. This ratio can be calculated for any territorial unit for any point in time, depending on the source of the population data. Roy and Ghosh have conducted several supervised machine learning techniques and identified population density as one of the most discriminatory factors contributing to infection and mortality rates [145]. Yazdani et al. reviewed 24 studies and found that the risk of COVID-19 transmission from sentinel cases to close contacts depends on the characteristics of the contact environment and Contact Environment Risk Score (CERS), which include the factor of crowdedness [146]. Velasco et al. studied 141 countries affected by COVID-19, and found that the total number of cases and deaths were significantly related to the population density [147]. On the other hand, Kawther and Maha analysed data associated with the spread of COVID-19 in Italy, Spain, and China during the period of these countries faced the highest numbers of COVID-19 infections in the world [148]. Statistical analyses of some characteristics of those countries confirmed that population density was one of the main variables contributing to the viral spread. This study used MATLAB software to build the COVID-19 fuzzy logic system to predict the spread of coronavirus. According to the findings, the confirmed number of COVID-19 cases and population density were positively correlated (*r* = 0.205). The output values from the fuzzy logic system also suggested that the rate of virus spread was higher in large cities, reflecting the effects of population density on COVID-19 infection. 

In Malaysia, a countrywide ecological study was conducted between 22 January 2021 and 4 February 2021 involving 51,476 active COVID-19 cases, and population density was found as an important factor in the spread of COVID-19 in Malaysia [149]. The study has generated Pearson’s correlation coefficients (*r*) with their corresponding 95% confidence intervals (CIs) to determine countrywide and regional-wise correlations between active COVID-19 cases and population density. From the countrywide perspective, Malaysia showed a statistically significant positive correlation between active COVID-19 cases and population density (*r* = 0.784; 95% CI 0.781, 0.787; *p* < 0.001). Based on the coefficient of determination (*R*^2^) values generated to determine the strengths of association between cases and population density, it confirmed that population density correlated with the spread of COVID-19 at the rate of 61.4% for the whole country.

### 7.2. Housing Condition

During the COVID-19 pandemic and the lockdown, it was a crucial period for people to stay home no matter the condition of the house or any kind of shelter they belong to. In this scenario, the poor housing conditions, particularly for low-income and minority ethnic populations, have become vulnerable to COVID-19 spread and infection [150]. Several studies have linked poor housing conditions with worse health outcomes and infectious diseases of COVID-19 spread [151,152]. Ahmad et al. conducted a cross-sectional analysis of county-level data in US among households with poor housing conditions, including conditions such as overcrowding, high housing cost, incomplete kitchen facilities, or incomplete plumbing facilities [151]. Due to a lack of proper plumbing and kitchen facilities in their houses, individuals would be forced to use communal facilities, increasing social contact. The study also found that with each 5% increase in percent of households with poor housing conditions, there was a 50% higher risk of COVID-19 incidence (IRR 1.50, 95% CI: 1.38–1.62) and a 42% higher risk of COVID-19 mortality (MRR 1.42, 95% CI: 1.25–1.61) across US counties. Overcrowding and poor plumbing may lead to recurrent exposure and potentially a higher viral inoculum, which had been connected to worse COVID-19 clinical outcomes. Varshney and Adalbert conducted bivariable and multivariable analyses using COVID-19 data from cities in Los Angeles County to determine the association between household overcrowding and risk of mortality from COVID-19 [152]. The bivariate regression revealed that overcrowded households were positively associated with COVID-19 deaths (standardized β = 0.863, *p*  <  0.001). The study concluded that residing in overcrowded households may be an important risk factor for COVID-19 mortality. This conclusion seems to be consistent with findings from a cross-sectional ecological analysis conducted by Daras et al. across 6789 small areas in England [153]. The study assessed the association between COVID-19 mortality in each area and living in overcrowded housing using the Estimates from multivariable Poisson regression models were used to derive a Small Area Vulnerability Index (SAVI). Daras et al. identified that the factor of living in overcrowded housing is independently associated with COVID-19 mortality. Counties with a higher number of households with inadequate housing showed an increased incidence of and mortality linked with COVID-19. These observations implied that targeted health policies to support individuals living in poor housing should be addressed in further efforts to reduce adverse outcomes related to COVID-19.

### 7.3. Economic Status

Many socioeconomic implications of the COVID-19 pandemic need to be explored further for better policymaking. This includes the importance of wealth or income level of society when it comes to the state of fighting against the spread of contagious diseases, particularly in many poor and large populated countries. Studies conducted to investigate the relationship between income level and COVID-19 infection rate or death rate are still limited and some have led to inconsistent results due to the adoption of different data sources. Most of the existing studies that used data aggregated at a geographical level could not estimate a direct association between individual income and COVID-19 outcomes [154]. Arceo-Gomez et al. has explored studies conducted in Sweden and Belgium that use individual-level data, which has revealed a negative association between income and COVID-19 mortality. Arceo-Gomez et al. through their research, had added value to the existing literature by providing evidence that low-income workers in Mexico suffered from higher hospitalization and fatality rates than their counterparts with higher wages [154]. On the other hand, Gong and Zhao studied the relationship between neighbourhood characteristics and the pandemic spread in Shenzhen, a Chinese megacity with many low-income rural migrants [155]. The study found that wealthier and larger neighbourhoods in Shenzhen were more likely to be infected in the first wave of the pandemic in early 2020. Specifically, one log point increases in average housing costs and property management fees are associated with a 127.1% increase and a 28.6% decrease in the odds ratio of infection, respectively. One log point increase in the number of buildings or apartments is associated with a 51.2–78.2% increase in the odds ratio of infection. This result contradicts the conventional belief that disadvantaged neighbourhoods are more susceptible to infection. This major reason is likely due to the richer people were more mobile and susceptible to infection at the beginning of the year 2020, introducing the virus into their neighbourhoods. 

Income inequality may increase the likelihood of infection because the most disadvantaged individuals must remain in the workforce to afford to live in a place where there are also many wealthy residents. One recent study by Oronce et al. revealed a significant association between increased state-level income inequality and the number of COVID-19 cases (*r* = 0.38, *p* = 0.006) and deaths (*r* = 0.44; *p* = 0.002) [156]. Similarly, another recent study by Liao and De Maio reported that an increase in a county’s income inequality correlated with an increase in COVID-19 incidence [157]. In their study, they found that a 1% increase in a county’s income inequality was associated with a 2.0% increase in COVID-19 incidence with an incidence risk ratio (RR) of 1.020 (95% CI, 1.012–1.027) and a 3.0% increase in mortality with a mortality risk ratio (RR) of 1.030 (95% CI, 1.012–1.047). Another study by Tan et al. also observed that income inequality within US counties measured by Gini coefficient was corresponded with more COVID-19 cases (Spearman ρ = 0.052; *p* < 0.001) and deaths (Spearman ρ = 0.134; *p* < 0.001) in the summer months of 2020 [158]. However, the association of income inequality and COVID-19 cases and deaths varied over time. They indicated that the adjusted association between the Gini coefficient and COVID-19 cases peaked in July and August 2020 (RR, 1.28; 95% CI, 1.22–1.28). These ecological analyses suggest a positive association between income inequality and COVID-19 incidences and deaths. High levels of income inequality may affect population health. Thus, to better understand the socioeconomic patterning of COVID-19 incidence and mortality, COVID-19 surveillance systems should account for county-level income inequality.

### 7.4. Education Level

According to the World Health Organization (WHO), literacy rate refers to the proportion of the adult population aged 15 years and over who is literate, referring to those who understand both read and write a short simple statement on his/her everyday life [159]. The growth of literacy rate or education level in a country has exposed the great importance of COVID-19 pandemic globally [160]. The effective and successful sharing of all relevant information to mitigate the fast spreading of COVID-19 requires the community’s understanding and appropriate response. Poorly educated people would not be able to understand the infections and the vaccination efforts, which played a vital role in overcoming this pandemic [161]. Rattay et al. highlighted education as an important determinant of COVID-19-related risk perception and knowledge which may influence protective behaviour [162]. This repeated cross-sectional online survey conducted during the pandemic in Germany from 3 March 2020, has concluded that education level should be considered in the development of risk communication strategies. This conclusion is consistent with a study conducted by Yoshikawa and Asaba, highlighting that a lower level of education influences the severity of COVID-19 among various populations [163]. Yoshikawa and Asaba claimed that their study is the first Mendelian randomization (MR) study that investigated the association between educational attainment and the risk of COVID-19 severity in the European population. They found that those with no qualification in the European population had a higher risk of severe COVID-19 than those with college or university degrees. Along with this, the hesitancy regarding the COVID-19 vaccine is also related to the studies on the association between education and COVID-19 related risk. Marzo et al. has conducted a descriptive cross-sectional study among 5260 participants in six Southeast Asian countries such as Indonesia, Malaysia, Myanmar, Philippines, Thailand, and Vietnam between February and May 2021 [164]. Based on the findings, about half (49.3%) showed their hesitancy to receive the COVID-19 vaccines and this hesitancy was found to be significantly associated with education levels. Participants with tertiary education were significantly more likely to agree that vaccines can effectively prevent and control COVID-19 compared to those with lower education who have expressed hesitancy in receiving COVID-19 vaccines. Populations with a higher level of education are typically more knowledgeable about vaccinations and the vaccination procedure [165], which increases their understanding of the risks and advantages of vaccination.

## 8. Host Factors

The global epidemic of SARS-CoV-2 poses the biggest threat to humanity due to its unfathomable magnitude. SARS-CoV-2 has rapidly spread throughout the community, infecting people of all ages, ethnicities, and gender. SARS-CoV-2 infection has been associated with specific host factors and comorbidities such as age, chronic lung diseases, hypertension, heart disease, kidney disease, immuno-deficiencies or metabolic disorders [166,167]. A study by Alam et al. showed that females were 1.69 times (AOR: 1.69; 95% CI: 1.52–1.87) more likely to become infected with COVID-19 than male participants, according to multivariate logistic regression. In contrast, a recent study concluded that females were less susceptible to COVID-19 than men due to differences in innate immunity, steroid hormones, and variables associated with sex chromosomes [168,169]. Progesterone, for example, can have many anti-inflammatory effects, mostly through inhibiting nuclear factor kappa beta and decreasing inflammatory cytokines such as, but not limited to, IL-12 and IL-10 [170]. Similarly, a population-based study in Iceland between 31 January to 31st March 2020 indicated that females and children under 10 years of age had a lower COVID-19 infection rate than males [171]. Another study in China demonstrated that men and women have the same prevalence, although men with COVID-19 are at a higher risk of poor outcomes and death, regardless of age [172]. It has been reported that levels of circulating ACE2 are higher in men than in women, as well as in individuals with diabetes or cardiovascular disease [173,174]. Previous studies found that increased ACE2 receptor protein expression in specific organs is linked with specific organ failures, as evidenced by clinical characteristics in COVID-19 patients [174,175]. 

SARS-CoV-2 can infect people of all ages; however, older individuals and those with pre-existing medical issues are more susceptible to infection and severe forms of COVID-19 [176]. The sensibility of the elderly could be explained by ACE2 overexpression or the high prevalence of comorbidities in this cohort [177]. A recent meta-analysis study indicated that patients older than 70 years have a higher infection risk (RR 1.65, 95% CI 1.50–1.81) and a higher risk for severe COVID-19 disease (RR 2.05, 95% CI 1.27 to 3.32) compared with patients younger than 70 years [178]. Based on a retrospective cohort study, COVID-19 infection risk was higher in 50–59 years (RR: 2.30, 95%CI: 1.65–3.27), 60–69 years (RR: 5.29, 95% CI: 3.76–7.46) and 70–79 years (RR: 3.03, 95% CI: 1.81–5.08) as compared to young adults aged 20–29 years. These findings confirmed that the elderly are more vulnerable to COVID-19 infection. The authors concluded that those aged 60–69 years might engage in more physical activities than those aged 70 years and older, which may result in closer contact with the index case for a longer duration [179]. Furthermore, the immunity of the elderly may be weaker than younger adults, rendering them more susceptible to COVID-19 infection. Similarly, a web-based survey also revealed that participants aged more than 60 years had a marginally higher risk (AOR: 1.36, 95% CI:1.16–1.60) than the younger participants [167]. They highlighted that the possible explanation for the increased risk of COVID-19 in the geriatric community is that most geriatric people reside in nursing facilities or old residences, which can facilitate rapid transmission [167]. Therefore, additional measures are required to protect the elderly from COVID-19 infection. 

Various studies have indicated that people with comorbidities were observed to have higher risks of COVID-19 infection than healthy people, irrespective of age and gender. One study conducted in Bangladesh reported that people with diabetes (AOR: 1.46, 95% CI:1.23–1.74, *p* < 0.001), liver disease (AOR: 9.39, 95% CI: 5.94–14.85, *p* < 0.001), kidney disease (AOR: 4.73, 95% CI: 3.38–6.61, *p* < 0.001) and heart disease (AOR: 2.08, 95% CI: 1.63–2.66, *p* < 0.001) had a significantly higher risk of COVID-19 infection when compared to those people who are not diagnosed with the disease. Moreover, comorbidities were associated with poorer clinical outcomes in COVID patients [180]. A meta-analysis revealed nine comorbidities with odds ratios of COVID-19 mortality up to 35 times greater than in the absence of comorbidities. The top four comorbidities were as follows: hypertension (OR: 34.73; 95% CI: 3.63–331.91; *p* = 0.002), diabetes (OR: 20.16; 95% CI: 5.55–73.18; *p* < 0.00001), cardiovascular disease (OR: 18.91; 95% CI: 2.88–124.38; *p* = 0.002), and chronic kidney disease (OR: 12.34; 95% CI: 9.90–15.39; *p* < 0.00001) [181]. People with hypertension or cardiovascular disease have a higher risk of COVID-19 infection and severe COVID-19 course potentially be associated with the use of angiotensin-converting enzyme (ACE) inhibitors and angiotensin receptor blockers (ARBs) for treatments. These medications are known to enhance the levels of ACE2 in numerous tissues, potentially increasing the power of SARS-CoV-2 invasion [182,183]. Furthermore, cardiac damage may occur during infection and treatment. Many anti-viral medications can cause cardiac insufficiency, arrhythmia, or other cardiovascular problems [184]. In addition to the general deficits in immunity (impaired phagocytosis, neutrophil chemotaxis, and T-cell activity) that predispose diabetics to infectious diseases such as SARS-CoV-2, diabetes may enhance virus entrance and multiplication [185]. Chronic obstructive pulmonary disease (COPD) and asthma have previously been linked to coronavirus infections such as SARS-CoV and MERS-CoV [186]. However, in the instance of COVID-19, it is unclear if COPD increases the infection rate, but it is known that people with COPD are more likely to present with severe symptoms. A recent study demonstrated that respiratory disease added only a modest increase in risk (OR: 6.69, 95% CI 1.06–42.26; *p* < 0.00001) of COVID-19 mortality. Although the mechanism promoting severe infection and poor outcomes in respiratory patients such as COPD is unknown, higher ACE2 expression has been linked to COPD and smoking [187,188]. Asthma is already known to increase susceptibility to viral infections due to a delayed innate antiviral immune response and decreased IFN production [186]. In a Korean nationwide cohort, it was found that people with nonallergic asthma had a greater risk of COVID-19 infection (AOR:1.34, 95% CI: 1.07–1.71) than those with allergic asthma (AOR: 1.06, 95% CI: 0.97–1.17) [189]. Despite the claim that asthma is a comorbidity in some cohorts, there is a distinct lack of evidence in the case of COVID-19 [190].

Recently, Popkin et al. found that overweight raised the incidence of COVID-19 by 44.0% (RR = 1.44; 95% CI: 1.08–1.92; *p* = 0.010), while those with obesity had nearly double the risk (RR = 1.97; 95% CI: 1.46–2.65; *p* = 0.0001) [191]. Obesity has also been proven to have detrimental effects on host immunity, raising the risk for infection susceptibility, severity, and unfavourable endpoints after infection, such as greater rates of hospitalization, intensive care unit admission, and mortality. Thus, obesity has been overlooked and one of the strongest COVID-19 risk factors. There is increasing evidence to indicate a strong association between obesity and COVID-19 infection [192,193]. Obesity is also associated with ACE2, adipose tissues express greater levels of ACE2, which might promote viral entry in adipocytes [194,195]. Therefore, a concern was raised if adipose tissue can function as a reservoir for the propagation of SARS-CoV-2 [196]. Furthermore, obesity is frequently accompanied by various comorbidities that raise the risk of COVID-19, including hypertension, diabetes, cardiovascular disease, and decreased lung function.

## 9. SARS-CoV-2 Variants

In the past few months, SARS-CoV-2 variants with unique spike protein mutations have emerged and impacted the epidemiological and clinical features of the COVID-19 pandemic. These mutated viral variants can increase transmission rates and/or increase reinfection rates, and impair the protection provided by neutralizing monoclonal antibodies and vaccination. These mutations can consequently allow SARS-CoV-2 to continue to spread in the face of increasing population immunity, while maintaining or enhancing its replication fitness [197]. Globally, as of 21 September 2022, there have been 610,393,563 confirmed cases of COVID-19, including 6,508,521 deaths, stated by the WHO [2]. The native SARS-CoV-2 virus, found in late 2019, evolved into many variants emerging across the world. It has been estimated that SARS-CoV-2 undergoes between 0.0004 and 0.002 mutations per nucleotide every year [198,199,200]. To prioritise monitoring and research, the WHO has classified these variants into three categories: variants of concern (VOCs), variants of interest (VOIs), and variants under monitoring (VUMs). The term “VOCs” refer to SARS-CoV-2 variants having changed phenotypic traits, such as greater transmissibility or virulence, or the capacity to avoid an immune response caused by natural infection or vaccination, or the ability to evade neutralization by monoclonal antibodies [201]. This indicates that multiple mutations in SARS-CoV-2 variants will significantly impact the virulence, transmission, and efficacy of diagnostics, vaccines, and antivirals. The European Centre for Disease Prevention and Control (ECDC) and the WHO had defined five VOCs as of December 2021: Alpha (B.1.1.7, first detected in September 2020 in the United Kingdom), Beta (B.1.351, first detected in May 2020 in South Africa), Gamma (P.1, first detected in November 2020 in Brazil), Delta (B.1.617.2, first detected in October 2020 in India), and Omicron (B.1.1.529, first detected in multiple countries in November 2021) [202]. All of these VOCs have amino acid alterations in the receptor binding domain (RBD) and N-terminal domain (NTD) of Spike protein, which are known to be the primary targets of neutralizing antibodies [203].

The Alpha (B.1.1.7) variant was detected for the first time in a sample obtained in the United Kingdom in September 2020 [204]. In the United Kingdom, Alpha was the predominant variant from December 2020 to May 2021, when it was replaced by Delta until December 2021, when it was surpassed by Omicron. According to a modelling study from the United Kingdom, the alpha VOC has a 43–90% (95% CI: 38–130) higher reproduction number than the original Wuhan strain [205]. By 16 March 2021, the Alpha variant had become dominant in 21 countries including the United Kingdom, the United States, Germany, and other European nations. In the United Kingdom, more than 275,000 cases of this variant were recorded by 16 March 2021. The Alpha variant first surfaced in the United States in November 2020, and the number of cases increased from 76 in 12 states on January 13 to 7501 in all 50 states by 23 March 2021 [206]. Hence, this variant has infected a significant proportion of the global population, although it is now regarded as de-escalated due to its low prevalence in the wake of Delta and Omicron. At least 22 mutations have been identified in this variant, including the N501Y mutation of the spike protein, which increases the affinity of RBD with ACE2, the P681H mutation, which is associated with an enhanced entry into the cell, and the D614G mutation in the spike protein, which also increases infectivity [207]. All these mutations led to increased transmissibility of this variant with an estimated 50% to 100% greater reproduction number [208]. Differences in clinical presentation between the Alpha variant and the wild-type SARS-CoV-2 were almost non-existent in hospitalized patients [209], despite some studies reporting increased mortality and severity in patients with this variant [210]. 

The Beta (B.1.351) variant first appeared in Nelson Mandela Bay, South Africa in October 2020, becoming the most prevalent form in the country until the emergence of Omicron, and has since spread to 115 countries [211]. The Beta variant contains nine spike protein mutations (L18F, D215G, K417N, D80A, D614G, E484K, N501Y, R246I, A701V). Three of them, K417N, E484K, and N501Y, have been linked to increased infectivity and mortality when compared to the Alpha variant, which only contains the N501Y mutation. All three are found in the RBD and enable the virus to adhere to human cells more easily while also increasing the binding affinity for the ACE2 receptor [212]. An ecological study in South Africa reported that Beta variants were found to be 41% (95% CI: 16–73) more transmissible and 53% (95% CI: 6–108) more fatal than non-Beta variants using the nationwide COVID-19 cases and SARS-CoV-2 sequences data [213]. This indicated that increased infection and mortality risks may increase the number of infections and critical patients. Pearson et al. estimated that the Beta variant of SARS-CoV-2 could be more transmissible than earlier circulating variants. It was discovered that the Beta variant accounted for approximately 40% of new SARS-CoV-2 infections, while the Alpha variant accounted for only 20% [214]. Pyke et al. reported rapid replication of the Beta variation, peaking at 48 h after infection with significantly greater levels of virus replication than the Alpha variant [215]. Apparently, the affinity of Beta RBD binding to ACE2 receptors is shown to be 2.7 times greater than that of the Alpha variant [216,217]. In terms of clinical implications of Beta variant, a higher risk of hospitalization, ICU admission, and mortality has been seen in contrast to Alpha and Gamma, which is similar to the Delta variation but less severe [218].

The Gamma (P.1) variant was discovered in Japan in December 2020, but it initially appeared in Brazil in late November 2020, developing from a local B.1.1.28 clade and displacing the parental lineage in less than 2 months, particularly during the second wave [219]. Gamma variants have at least 23 distinctive mutations and, like Beta and Gamma variants, also have K417N/T, E484K, and N501Y mutations, which are responsible for increased infectivity [220]. A recent study from southern Brazil revealed that the Gamma variant resulted in higher viral loads and 1.4–2.2 times higher transmissibility than previous strains, particularly in young patients aged 20–59 years old without pre-existing risk conditions (no gender difference) [221]. While this variant has spread to other countries, particularly in South and Central America, the Gamma variant has become endemic in Brazil, particularly in the State of Amazonas, where many Gamma Plus variants have evolved. These variants contain additional S protein mutations than the parental P.1, making them more transmissible [222]. There is no evidence of a link between Gamma variant infections and the severity of COVID-19 symptoms. However, its rapid spread and socioeconomic and public health restrictions may explain the increased number of cases and deaths in Brazil during the second wave [223].

The Delta variant (B.1.617.2) was first identified in India in December 2020, competing with the previously identified B.1.1.7 (Alpha) [224]. The American Society for Microbiology reported that the Delta variant accounts for 83% of cases in the United States and 90% in the United Kingdom. They also claim a 40–60% rise in transmissibility when compared to the alpha variant, which was twice as contagious as the original Wuhan strain [225]. The advent of this variant has resulted in the number of COVID-19 infections in India drastically rising, leading to a massive epidemic situation accompanied by numerous record-breaking cases and deaths. More than 400,000 new cases and 4000 deaths were reported in one day, making it the first country in the world to be in this situation. As of 14 March 2022, the National Genomic Surveillance of SARS-CoV-2 variants of Indonesia has revealed that the Delta variant was the most prevalent circulating variant in Indonesia (50.9%), followed by Omicron (48.5%), Alpha (0.50%), and Beta variants (0.13%) [226]. Furthermore, this variant has been demonstrated to have a 108% increase in hospitalization risk, a 235% increase in ICU admission, and a 133% increase in mortality risk when compared to the original variant [227]. These data could be attributed to the higher viral load of the Delta variant compared to other variants, being more transmissible among humans (particularly via aerosols) without directly affecting mortality [228]. Several studies observed that increased infectivity of the Delta variant may be due to its 10 notable spike protein mutations: T19R, 156del, 157del, L452R, G142D, D614G, D950N, P681R, R158G and T478K [224]. Generally, the mutations identified in the Delta variant show an improved adaptation to human ACE2, and the ability to broaden their host ranges com-pared to the original SARS-CoV-2 [229]. It has been demonstrated that the L452R mutation allowed this variant to bind to human ACE2 with higher affinity than other variants, and evaded the attack of CD8 T cells, which are the cells that eradicate the virus [230]. A study indicated that the P681R mutation in the Delta variant enhanced cell-surface-mediated virus entry to the host cell compared to other variants without such mutation [231]. In terms of clinical manifestations, the Delta variant appears to have a shorter time interval between disease onset and hospitalization than the original SARS-CoV-2, as well as significant changes in its haematological profile. Moreover, individuals with COVID-19 infected with the Delta variant had higher hospitalization or emergency care attendance risks than those infected with the Alpha variant, as well as an enhanced transmissibility [232,233]. According to one Scottish study, the delta variant doubled the risk of hospitalization when compared to the Alpha variant. The study also found that the Delta variant was more widespread among their younger and affluent populations [234]. Another study revealed that the Delta variant was the dominant strain and has been spread rapidly in primary and secondary schools in the United Kingdom [235]. High transmission rates in the Delta variant can result in high mutation rates and the emergence of new strains. Therefore, the experts are collectively monitoring and surveilling the Delta Plus and other subtypes [236]. 

The Omicron variant (B.1.1.529) was first discovered in Botswana, South Africa in November 2021 and reported to the WHO on 24 November, where it was designated as VOC two days later [237]. This variant contains more than 30 spike protein mutations, 15 of which impact the RBD, which is essential for the interaction with ACE-2 [238]. These changes facilitated the rapid spread of this variant, affecting almost 2000 persons in 57 countries within two weeks. The Omicron variant had been found in more than 188 countries as of 31 March 2022 and had already become the dominant strain on a worldwide scale. From 19 August to 19 September 2022, a total of 120,617 SARS-CoV-2 sequences were shared globally via the GISAID (Global Initiative on Sharing All Influenza Data) database. Among them, 119 458 sequences belonged to the Omicron variant, which accounted for 99.0% of all sequences reported globally in the previous 30 days [239]. Some hypotheses suggest that the emergence of this variant occurred in an immunocompromised individual infected with the Human Immunodeficiency Virus (HIV) or through recombination with the human common cold coronavirus, whereas other studies suggest a possible animal (mouse) origin of this variant [240,241]. Other plausible explanations include the variant possible circulated and evolved in a remote population under intense evolutionary pressure, immunization inequality or inadequate vaccine coverage, which provided an ideal evolutionary scenario for Omicron [242]. Many studies have indicated that the Omicron variant is more transmissible than the Delta variant. A France study reported that the Omicron variant was 105% (95% CI: 96–114%) more transmissible than the Delta variant [243]. Another recent study in Denmark found that the effective (instantaneous) reproduction number of the Omicron variant was 3.19 (95% CI: 2.82–3.61) times greater than that of the Delta variant [244]. Regarding its clinical manifestations, it appears that the Omicron variant is associated with a milder clinical presentation than the Delta variant. However, the Omicron variant has surpassed the Delta variant in multiple countries. It is difficult to determine the underlying severity of this variant because, in comparison to Delta, the worldwide percentage of vaccinated people may explain some of the discrepancies [245]. However, there are some concerning factors behind why Omicron is classified as a VOC. It is known, for example, that vaccinated patients and convalescent individuals who have previously been infected with other variants have significantly lower sera neutralization titres against the Omicron variant. Fortunately, patients who received the third vaccine dose or who had previously been infected with the Delta variant may have a stronger antibody response to the Omicron variant [246,247]. Finally, there has been some debate about Omicron’s role in the pandemic. Additionally, as the Omicron variant infection leads to less severe COVID-19, it may help to halt the pandemic [248]. Despite this trend, more variants are likely to emerge in the future, as humans and SARS-CoV-2 strains are supposed to coexist.

## 10. COVID-19 Testing Availability

On 16 March 2020, the WHO conveyed a simple message to all countries—test, test, test—which indicated that COVID-19 testing is the key to controlling virus transmission (WHO, 2020). This is especially true for SARS-CoV-2, as asymptomatic and presymptomatic people play an important role in spreading the virus [249]. The WHO also urged all countries to increase their testing programs as the best method to halt the spread of the coronavirus pandemic, as well as industries to boost the production of key equipment to address urgent shortages [250]. The presence of SARS-CoV-2 is commonly tested for one of two reasons. First, a symptomatic patient may be tested in order to guide their clinical treatment. Such diagnostic testing is normally performed in a well-controlled clinical setting, and the test results are usually interpreted in conjunction with the patient’s history and symptoms. Second, testing may be carried out to identify infectious individuals in a population, who are subsequently isolated to pre-vent further transmission. Such screening aims to improve public health outcomes by reducing viral transmission in a community. Asymptomatic individuals might be tested, and testing on a massive scale may be required. These two types of testing have different priorities and requirements, and a test that is beneficial in clinical diagnosis may be unsuitable for population-scale screening. Reports have indicated that countries with high testing rates were able to effectively reduce SARS-CoV-2 transmission during the early stages of the pandemic, reinforcing the proposal that screening could aid in viral transmission management [251].

When an epidemic re-emerges, it is critical to reinforce public health measures such as physical separation, contact tracing, testing, and quarantine [252]. In China, COVID-19 mass testing was one of the public health measures for infection elimination in the second phase of the epidemic. Mass testing is essential to identify infected individuals and COVID-19 hotspots and to assist in surveillance, contact tracing, and early quarantine to prevent further transmission [253]. For example, a city-wide mass screening in China was conducted in 4090 testing locations in Qingdao and suburban areas to break the COVID-19 transmission. Within five days, 10.9 million people were tested, and the outbreak was contained without a lockdown [254]. Similarly, as a measure of COVID-19 control, New Zealand has conducted extensive testing, including testing asymptomatic individuals early in an outbreak and testing populations at high risk from certain localities [255]. Early mass testing facilitates rapid detection and isolation of new cases before epidemiological inquiry yields indications to prevent future transmission; hence, all contacts can be detected sooner and isolated for the incubation period. The infection rate and burden of future isolation will be significantly lowered by minimizing the possibility of subsequent transmission. However, such massive testing initiatives can quickly strain supply chains and infrastructure, and many countries have experienced reagent shortages [256] that prevented them from implementing effective, widespread testing. Consequently, the testing of healthcare providers and symptomatic persons is frequently prioritized above the testing of asymptomatic individuals. Competition for reagents has frequently mirrored global health inequities, and there are large differences in testing rates between countries [257]. 

The strategy of tracking, testing, and treatment has become synonymous with COVID-19 pandemic control. Understanding COVID-19 epidemiology in each region of the world is critical for policymaking; yet, this is not always possible due to restricted availability of tests, inadequate laboratory facilities, insufficient employees, and overburdened health systems, especially in low and middle-income countries (LMICs). Furthermore, the so-called COVID-19 “infodemic” has aided in the spread of anti-science sentiments and has had a negative impact on testing and vaccine uptake globally [258]. This pandemic has been marked by alarming inequalities. The LMICs have failed to gain access to existing tools to mitigate COVID-19’s impacts on their fragile health systems [259]. From 16 September 2021 to 16 September 2022, an average of 0.69 tests were performed every day for every 1000 people worldwide. However, this average conceals the disparity: high-income countries conducted 4.16 tests per 1000 people on average, while low-income countries conducted only 0.06 tests per 1000 people. Till now, only 21.2% of tests administered worldwide have been used in low- and lower-middle-income countries, despite these countries accounting for 50.8% of the global population [260]. The pandemic has revealed the insufficiency of global supply chains in ensuring that countries have access to the tools necessary to combat COVID-19. The reliance on a few of diagnostics manufacturers has led to unfair competition for scarce resources and jeopardized the ability of LMICs to expand testing coverage. For example, Malawi, one of the LMICs, struggled with testing since the first reported case on 2 April 2020, due to insufficient capacity, unskilled laboratory personnel, insufficient funds, and a lack of policies. Malawi only managed to analyse around 21,500 samples as of 15 July 2020, representing approximately 0.1% of the overall population with a total population of 18 million. The screening and testing in Malawi were relatively lower than in other African countries [261]. 

Besides that, Ecuador, which has one of the highest excess mortality rates in the world, had a comparatively low number of confirmed COVID-19 cases, which could have been influenced by limited data availability for decision-making due to poor laboratory capability for testing. Ecuador, similar to many LMICs experiencing COVID-19, does not produce laboratory supplies, and disruptions in global transit diminished imports while the national lockdown delayed customs clearance. Swabs, reagents, and RNA extraction kits have been in limited supply, and the prices of necessary personal protective equipment required to collect and process samples, such as N95 masks, have risen, resulting in more expensive testing. Furthermore, not all laboratories are capable of processing high numbers of samples due to manpower shortages, including technicians and administrative assistants, as well as a lack of laboratory equipment and computing resources to analyse data and aid in workflow management [262]. This is why Ecuador may be under-testing its population, with only 46,209 tests per million population (pmp) as of 18 January 2021, compared to 216,382 pmp in Uruguay Center for Systems Science and Engineering, 2021.

According to preliminary research, COVID-19 testing barriers exist along multiple and intersecting dimensions. Testing barriers might arise as a result of cost of testing, testing centre locations, testing centre hours, communication strategies, inaccessible environments, and decisions about how testing is allocated [263]. For instance, COVID-19 tests are consistently free in public facilities in India and Zambia. In Kenya and the Philippines, this is not the case. In the Philippines, the average cost of a diagnostic test is USD 55, or two days’ pay for the average wage earner. In Kenya, the average price was equivalent to one day’s salary for the ordinary person, or USD 11. This equates to six full days’ pay for the 37 percent of Kenyans who live in extreme poverty (earning less than USD 1.90 per day) and may be particularly vulnerable to transmission [264]. Additionally, several studies indicated that the availability of testing sites is a process barrier for testing [265,266]. Clusters research in New York City revealed a shortage of testing centres in lower socioeconomic communities, which had a larger proportion of positive cases and more severe illnesses than higher socioeconomic communities [266]. Furthermore, Maxmen et al. reported that rural and distant areas had even less access to testing. Similarly, as the distance to testing sites increased in rural areas, testing accessibility decreased [267]. In Australia, testing availability and timeliness have also been highlighted as barriers to testing in remote communities [268].

Antigen rapid diagnostic tests (Ag-RDTs) are the most promising techniques for SARS-CoV-2 testing scale-up among the numerous types of tests available. Ag-RDTs deliver good results in a short period of time and do not necessitate sophisticated laboratory infrastructure or expert employees. This allows for decentralized testing; therefore, Ag-RDTs could potentially expedite COVID-19 detection and reaction at scale in LMICs, assisting in pandemic control [269]. The role of diagnostics has developed and will continue to evolve during the COVID-19 crisis. Increased access to testing will improve surveillance, create real-world vaccination efficacy data, guide strategies, and permit the monitoring of emerging variants. As outpatient clinical trials continue, increased access to tests will aid in demand forecasting and treatment adoption [259]. Containing SARS-CoV-2 transmission, on the other hand, is a global challenge, and global responsibility for sustainable testing must be recognized.

## 11. Challenges and Recommendations

Determining the factors of COVID-19 infections can be challenging as infections are not always caused by a sole component but by multiple factors. Many other external or unseen factors can cause COVID-19 infections. The emergence of a new virus variant is possible due to slow “herd immunity” progress. This is attributed by the high number of unvaccinated people, slow vaccination programs in certain countries, and a surge in virus transmission. Historical data of a person, especially on travel places and contact tracing, can be laborious without a proper strategy by lawmakers. In addition, undisclosed health info of a person and their current condition can result in undetected infections. Other than that, the prediction of when the virus will mutate and merge into a new variant is also one of the challenges to end the pandemic. Furthermore, the methods for detecting the virus are only through commercial COVID-19 tests such as RTK and RT-PCR using saliva or nasal samples. Since the virus is transmitted via airborne, it is important to study the viability of the virus in the air. However, the SARS-CoV-2 virus viability in the air sample is still controversial since several studies demonstrated that only a small percentage of the air samples obtained from the hospital rooms of COVID-19 patients are detected with the virus when they are tested using RT-PCR and virus culture [270,271,272]. Nevertheless, these laboratory-based detections of SARS-CoV-2 in air samples certainly imposes a severe challenge to curb the infection risk. Therefore, detection of the virus using a biosensor was proposed to provide a more rapid, portable and point-of-care system [273]. 

To overcome the issues and challenges, further research in the medical field on various aspects, such as technology on virus detection can be further discovered. A method to assess aerosol dynamics in current COVID-19 pandemic during dental restoration and cleaning procedures using a real-time sensor network has been conducted [274]. Currently, most studies in COVID-19 scope focus on the effectiveness of commercial detection and prevention methods. The utilization of machine learning to predict new COVID-19 variant or other zoonotic diseases can be explored since currently, there are study that utilizes machine learning to predict new COVID-19 emergence [275].

The COVID-19 pandemic has exposed the vast differences in health outcomes resulting from income inequality. In addition, income inequality may worsen the impact of the COVID-19 outbreak since low-income individuals are more likely to work in essential occupations with high exposure risk and have less access to healthcare. Therefore, targeted interventions should be concentrated on areas of income inequality in order to both flatten the curve and reduce the burden of inequality. Personal protective equipment distribution, increased COVID-19 testing, educational campaigns, additional information on COVID-19 non-pharmaceutical therapies, and ultimately, enhancing vaccination acceptance among individuals at highest risk of exposure are all potential targeted interventions.

## 12. Conclusions

Precise knowledge of the COVID-19 infection factors can effectively control the disease and predict the infection rate. However, the predictive analysis and control mechanism must be adaptive to the dynamic situations based on several factors and policies implemented at a particular period. Therefore, this paper has comprehensively analysed the different factors that affect the COVID-19 infection rate or risk. For each factor, quantitative and qualitative analyses were provided to enable further profiling and prediction of COVID-19 infection risk, either by policymakers or researchers. Based on the reviewed factors and their associated analyses, challenges of accurate prediction of COVID-19 infection risk were outlined, with the most challenging aspect as the multi-dimensional consideration that is dynamic based on different situations and conditions. Lastly, the review has articulated the advancement of using artificial intelligence-based methods to implement adaptive learning based on dynamic situations and to carry out prediction of COVID-19 infection based on multi-parameters input.

## Figures and Tables

**Figure 1 ijerph-19-12997-f001:**
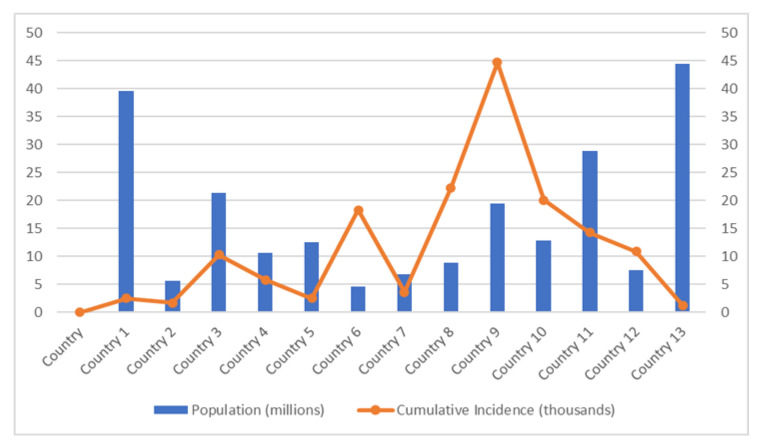
Cumulative COVID-19 cases in different populations.

**Figure 2 ijerph-19-12997-f002:**
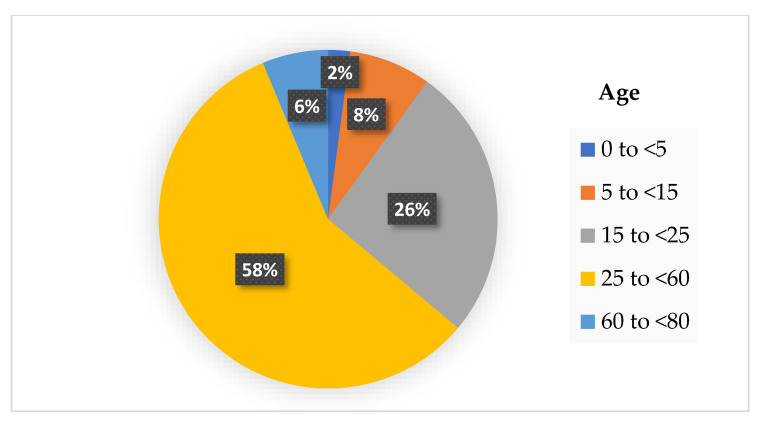
The percentage of COVID-19 infection cases based on age difference with reduced mobility.

**Figure 3 ijerph-19-12997-f003:**
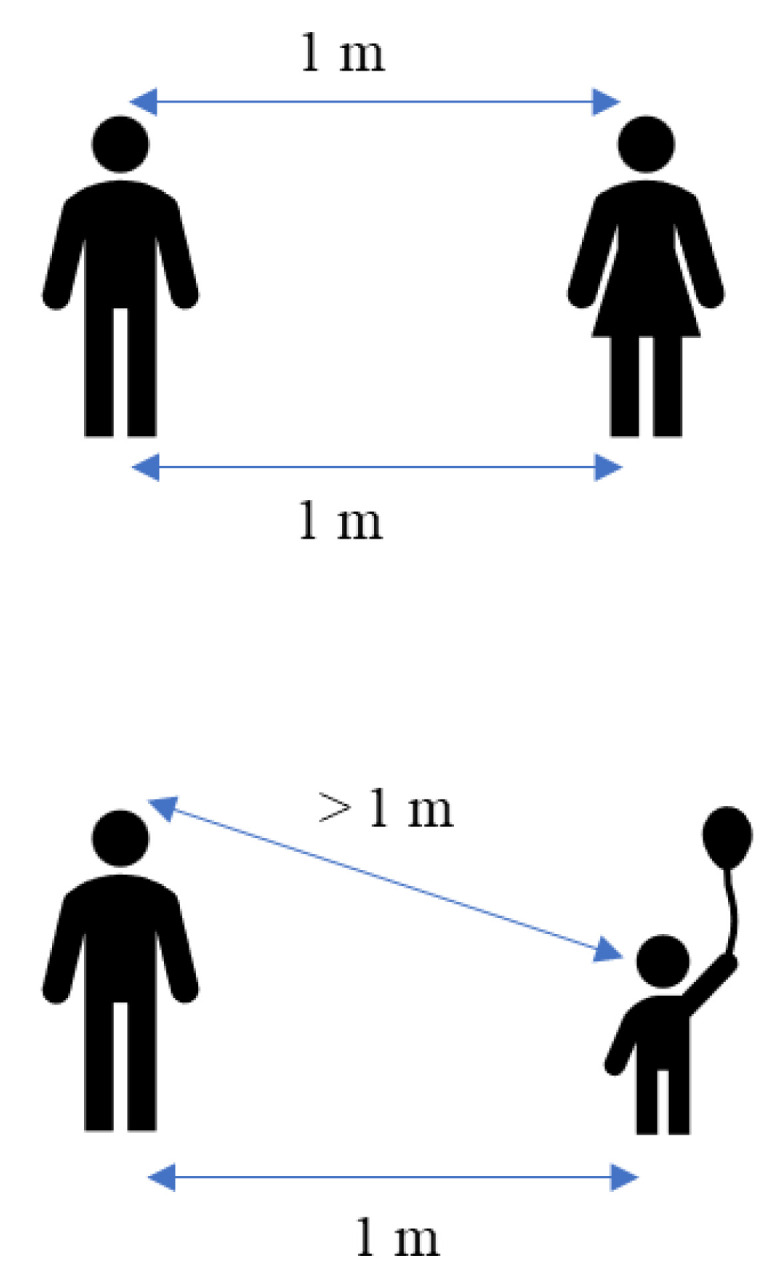
Theoretical of different transmission rates with the various height.

**Figure 4 ijerph-19-12997-f004:**
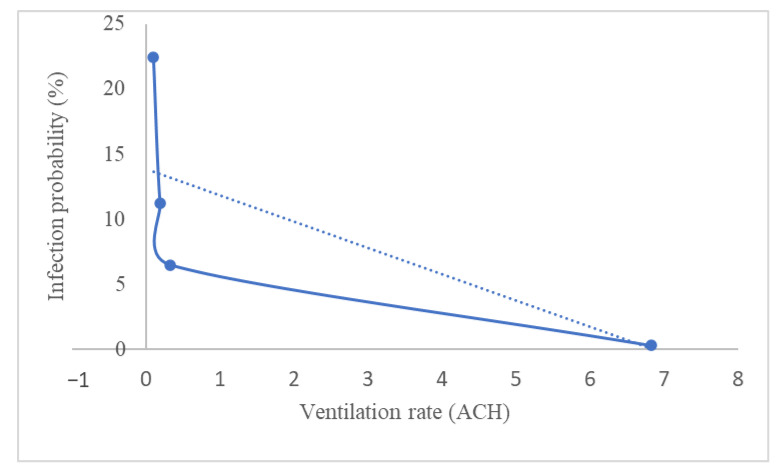
Infection probability based on the ventilation rate in an average of 1 h in a classroom, with the students wearing a mask (ACH: air changes per hour).

**Figure 5 ijerph-19-12997-f005:**
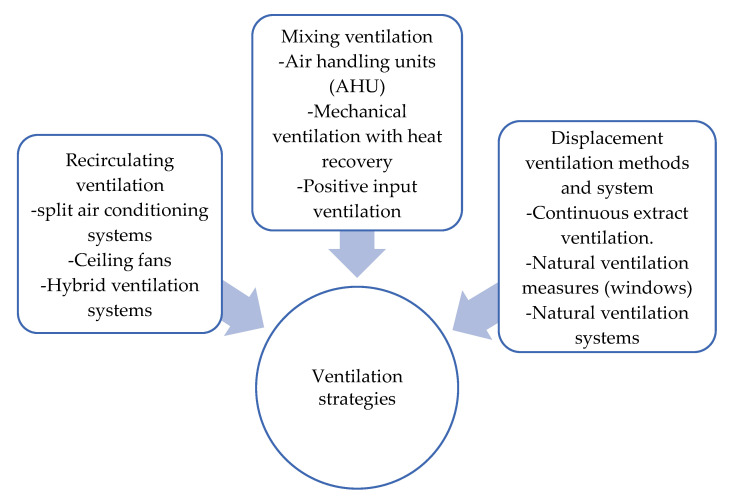
Ventilation strategies by airflow dynamics.

**Figure 6 ijerph-19-12997-f006:**
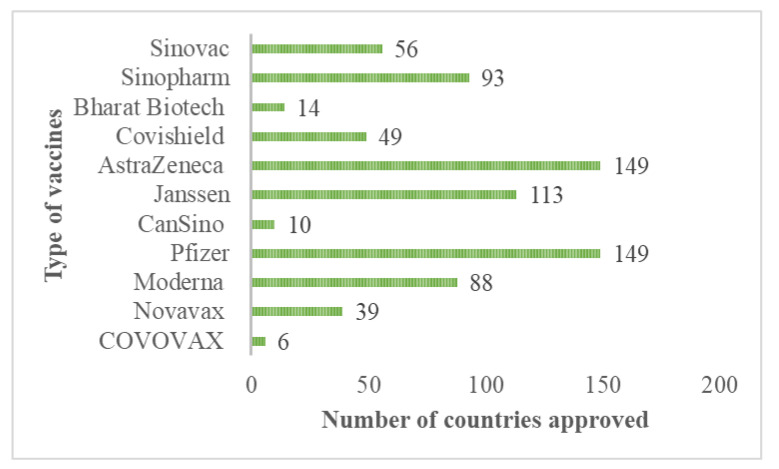
Type of vaccines approved by certain countries.

**Table 1 ijerph-19-12997-t001:** Materials and protection level for different types of face mask.

Type of Facemask	Material	Protection Level	BFE (%)
Surgical	Non-woven fabrics with a multi-layered structure consisting of a leak-proof layer, a high-density filter layer, and a direct contact skin layer	Prevent cross-contamination between respiratory particles of a wearer and body fluid of patients	>99
Respirator (N95)	Multiple layers of non-woven fabric, often composed of polypropylene, and pre-filtration layer	Filter contaminants, bacteria	>99
Cloth	Cotton/cloth	Mitigate aerosol dispersal, reduced transmission through droplets	96

**Table 2 ijerph-19-12997-t002:** Details of facemask specifications used in Chiera et al. study [31].

Mask ID	Mask Type (Manufacturer Claim)	Filter Material	No. of Layers	DP	BFE (%)	Fibre Structure	Filtering Area (cm^2^)	Total Mask Size (cm^2^)	Fitting System	Nose Piece
CM-1	Community, reusable	100% cotton	2	315	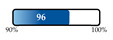	Woven fabric	329	393	Ear loops	Metal wire
CM-2	Community, reusable	92% cotton, 8% PU	2	56	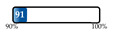	Knitted fabric	225	259	Ear loops	None
CM-3	Community, reusable	PP	3 (SSS)	10	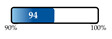	Non-woven	308	347	Ear loops	None
CM-4	Community, single-use	PP	1 (S)	7	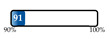	Non-woven	356	396	Ear loops	None
SM-1	Surgical Type I, single-use	PP	3 (SMS)	28	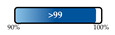	Non-woven	188	271	Ear loops	Metal wire
SM-2	Surgical Type I, single-use	PP	3 (SSS)	77	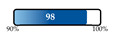	Non-woven	207	286	Ear loops	Metal wire
SM-3	Surgical Type I, single-use	PP	3 (SMS)	35	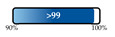	Non-woven	200	277	Ear loops	Metal wire
SM-4	Surgical Type IIR, single-use	PP	3 (SMS)	30	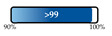	Non-woven	200	272	Ear loops	Metal wire
FP-1	FFP2 respirator	PP	3 (SMS)	58	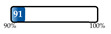	Non-woven	192	252	Ear loops	Metal wire
FP-2	FFP2 respirator	PP	3 (SMS)	53	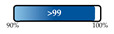	Non-woven	173	255	Head loop	Metal wire + foam

The -1, -2, -3, or -4 label at the end of each mask ID are to differentiate different design and material of face mask from the same type. DP: breathability according to EN 14683; BFE: bacterial filtration efficiency according to EN 14,683 measured using a bacterial aerosol having a mean particle size of 3.0 ± 0.3 μm; FFP: filtering face piece; PU: polyurethane; PP: polypropylene; S: spunbonded; M: meltblown.

**Table 3 ijerph-19-12997-t003:** Exemption of face mask by CDC.

Type of Exemptions	Factors of Exemptions
Exempted	A person with a disability who, due to the disability, would be physically unable to take off a mask on their own if breathing becomes obstructed. Examples includes a person with impaired motor skills, quadriplegia, or limb restrictions. Inability to comprehend the necessity of removing a mask if breathing becomes difficult due to an intellectual, developmental, cognitive, or psychiatric disability.
May be exempted	Wearing a mask over the mouth and nose would prevent breathing or cause respiratory discomfort. Persons with intermittent respiratory distress illnesses, such as asthma, are unlikely to be excused from this rule because they can typically wear masks without risk. require the use of an assistive device that restricts the user from wearing a mask and utilizing the device at the same time, such as one for mobility or communication. If the device is only intermittently used and the person is capable of removing the mask on their own, a mask must be worn at all times while the person is not using the device. Severe mental illness or a severe sensory impairment that, if wearing a mask was required, would immediately potentially harm the wearer or others. This exemption would not apply to people who feel uncomfortable or anxious while wearing a mask and no immediate danger appears to exist.

## Data Availability

Not applicable.

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
