# Peer review of "Prerequisite for COVID-19 Prediction: A Review on Factors Affecting the Infection Rate"

_ijerph, 2022, doi:10.3390/ijerph192012997_

Round 1

Reviewer 1 Report

In this review article, Tang et al., present the various elements with the potential to affect the dynamics of COVID-19 in the world, taking into account the lack of clarity in the literature about the real effects of each measure on the case series of infection. The review is of high quality and covers different aspects of the possible mechanisms of transmission/control of the pandemic, but it must also present the controversies on the topics, given that the scientific discussion must encompass the opposites in the search for possible consensus, and due to the real lack of knowledge about the factors linked to the pandemic. Thus, some suggestions were made to the authors:

1)      Introduction:

-          Although the bat-origin is a plausible hypothesis, it´s important to emphasize that the matter is not settled as SARS-CoV-2 samples were not obtained in nature after diverse comprehensive wild life probing.

-          Besides the clinical manifestations already described, it´s interesting to pinpoint Long COVID syndrome as a latter potential complication.

2)      Face masks: briefly discuss the groups not recommended for the use of masks and the psychosocial inconveniences of their use in society in general.

3)      Physical distancing:

- Two topics with the same name (4.1 and 4.2)

- The question of the effectiveness of distancing measures was one of the most debated, as they seem to have a different impact on the number of cases. For example, some studies indicated that an important part of the contacts was in the home environment, not outside. In the same way, transmission in a school environment did not prove to be exactly important. In this way, it has to be also questioned that many non-empirical mathematical models point to rates that are often excessive in relation to reality. These are elements that can be pointed out in the review.

- Since the authors comment on the transmissibility of droplets/aerosols, it would be interesting to briefly comment on the issue of transmissibility of surfaces as well, a fact that has not yet been definitively demonstrated.

4)      Vaccines: it would be interesting to comment that the circulation of the virus continues, especially with the Omicron variant, despite high rates of vaccination around the world. Despite this, the vaccine remained effective in containing severe cases/deaths. Changes in vaccine composition have been implemented (bivalent) with still unknown actual impact.

5)      The availability of testing for SARS is a highly relevant factor for early case identification and isolation/quarantine of contacts. I believe that an approach to the matter would be necessary.

Reviewer 2 Report

The study by Tang et al., “Prerequisite for COVID-19 prediction: A review on factors affecting the infection rate” reports a comprehensive review of the different factors influencing the COVID-19 infection rate. Authors suggest that factors such as physical distance, ventilation, face masks, meteorological factor, socioeconomic factor, and vaccination impact the COVID-19 risk directly or indirectly. Overall, it is a very interesting review of the factors affecting COVID-19 infection rate or risk and their multi-dimensional consideration. The manuscript is also well written; however, I have the following minor comments:

1. Authors should discuss the gender differences in vaccine hesitancy as it is a crucial factor in vaccine coverage, especially in developing countries. Additional information about vaccine hesitancy in recovered patients as well as for booster doses would add value to this study.

2. In lines 599-601, the sentence "According to World Health Organization (WHO), literacy rate refers to….." please add references to this statement.

3. In line 238, please check the sentence and correct it.

Reviewer 3 Report

I think it is very intersting review regarding COVID-19 .

I would recommend to add some data regarding host factors as well as the different variants of the coronaviruses .  Some of the studies that were published highlighted this issues (For example, among dialysis population,  despite homogeunose nature of the patients ,  additional factors such as different variant strains of SARS-CoV-2  are critical for infection and outcomes of the infection itself. 

some minor changes are also required (add abbreviations to the Tables etc) ,  revise reference list.  

Reviewer 4 Report

Title:

“Prerequisite for COVID-19 prediction: A review on factors affecting the Infection rate”

Review

The manuscript has reviewed many factors that have an effect on the COVID-19 infection rate, considering SARS-CoV-2 transmission and its different components. The issue is important. The objective of this review is to improve the manuscript. Some suggestions may be addressed:

1. It may be convenient to more detail on the transmission of COVID-19, including the estimated percent of asymptomatic COVID-19 cases, the role of super spreaders, mass or social gathering events, and the rapid diffusion of the virus (1-2).    

2. The places where the virus transmission is more probable and more outbreaks have occurred could be described, including nursing homes, hospitals, work centers, leisure places, schools,  cruise ships, and others (3-5).

3. Detection of SARS-CoV-2 in the air (Page 12 line 642).The paragraphs about the detection of SARS-CoV-2 in samples of air may be confused. The virus had been detected in samples of air in hospitals and rooms with COVID-19 patients (6-8). In addition, virus viability in air samples is controversial (9-10)   

4. With respect to COVID-19 vaccination may be useful to mention the different vaccines and their efficacy in the general population (11). On the other hand, an indication of if the actual vaccines could stop the transmission of the virus as well as the duration of the protection may useful (12-13).

5. Reference number 108 could be not correct. We think that this article has been published in Science (14).

6. The reference number 97 is indicated in the text as Eva et al (line 563). The first author is Eva O Arceo-Gomez. The corrected cite is Arceo-Gomez et al.

7. The references of the manuscript need to follow the guideline of the journal.   

References

1.Whaley CM, Cantor J, Pera M, Jena AB. Assessing the association between social gatherings and COVID-19 risk using birthdays. JAMA Intern Med. 2021 Aug 1;181(8):1090-1099. doi: 10.1001/jamainternmed.2021.2915.

2.Domènech-Montoliu S, Pac-Sa MR, Vidal-Utrillas P, Latorre-Poveda M, Del Rio-González A, Ferrando-Rubert S, et al. "Mass gathering events and COVID-19 transmission in Borriana (Spain): A retrospective cohort study". PLoS One. 2021 Aug 26;16(8):e0256747. doi: 10.1371/journal.pone.0256747.

3.Hongoh V, Maybury D, Levesque J, Fazil A, Otten A, Turgeon P, et al. Decision analysis support for evaluating transmission risk of COVID-19 in places where people gather. Can Commun Dis Rep. 2021 Nov 10;47(11):446-460. doi: 10.14745/ccdr.v47i11a02.

4. Brown KA, Jones A, Daneman N, Chan AK, Schwartz KL, Garber GE, et al. Association between nursing home crowding and COVID-19 infection and mortality in Ontario, Canada. JAMA Intern Med. 2021 Feb 1;181(2):229-236. doi: 10.1001/jamainternmed.2020.6466.

5.Dahl E. Coronavirus (Covid-19) outbreak on the cruise ship Diamond Princess. Int Marit Health. 2020;71(1):5-8. doi: 10.5603/MH.2020.0003.

6. Kenarkoohi A, Noorimotlagh Z, Falahi S, Amarloei A, Mirzaee SA, Pakzad I, et al. Hospital indoor air quality monitoring for the detection of SARS-CoV-2 (COVID-19) virus. Sci Total Environ. 2020 Dec 15;748:141324. doi: 10.1016/j.scitotenv.2020.141324.

7. Chia PY, Coleman KK, Tan YK, Ong SWX, Gum M, Lau SK, et al. Detection of air and surface contamination by SARS-CoV-2 in hospital rooms of infected patients. Nat Commun. 2020 May 29;11(1):2800. doi: 10.1038/s41467-020-16670-2.

8. Breshears LE, Nguyen BT, Mata Robles S, Wu L, Yoon JY. Biosensor detection of airborne respiratory viruses such as SARS-CoV-2. SLAS Technol. 2022 Feb;27(1):4-17. doi: 10.1016/j.slast.2021.12.004.

9. Lednicky JA, Lauzard M, Fan ZH, Jutla A, Tilly TB, Gangwar M, et al. Viable SARS-CoV-2 in the air of a hospital room with COVID-19 patients. Int J Infect Dis. 2020 Nov;100:476-482. doi: 10.1016/j.ijid.2020.09.025.

10. Ong SWX, Tan YK, Coleman KK, Tan BH, Leo YS, Wang DL, et al. Lack of viable severe acute respiratory coronavirus virus 2 (SARS-CoV-2) among PCR-positive air samples from hospital rooms and community isolation facilities. Infect Control Hosp Epidemiol. 2021 Nov;42(11):1327-1332. doi: 10.1017/ice.2021.8

11. Wang K, Wang L, Li M, Xie B, He L, Wang M, et al. Real-Word Effectiveness of global COVID-19 vaccines against SARS-CoV-2 variants: A systematic review and meta-analysis. Front Med (Lausanne). 2022 May 19;9:820544. doi: 10.3389/fmed.2022.820544.

12.Ioannou P, Karakonstantis S, Astrinaki E, Saplamidou S, Vitsaxaki E, Hamilos G, et al. Transmission of SARS-CoV-2 variant B.1.1.7 among vaccinated health care workers. Infect Dis (Lond). 2021 Nov;53(11):876-879. doi: 10.1080/23744235.2021.1945139.

13.Hsu L, Hurraß J, Kossow A, Klobucnik J, Nießen J, Wiesmüller GA, et al. Breakthrough infections with the SARS-CoV-2 Delta variant: vaccinations halved transmission risk. Public Health. 2022 Mar;204:40-42. doi: 10.1016/j.puhe.2022.01.005.

14.Obermeyer F, Jankowiak M, Barkas N, Schaffner SF, Pyle JD, Yurkovetskiy L, et al. Analysis of 6.4 million SARS-CoV-2 genomes identifies mutations associated with fitness. Science. 2022 Jun 17;376(6599):1327-1332. doi: 10.1126/science.abm1208.
